# Isothermal amplification and fluorescent detection of SARS-CoV-2 and SARS-CoV-2 variant virus in nasopharyngeal swabs

**Les Jones[1,2], Abhijeet Bakre[1,2], Hemant Naikare[1,3], Ravindra Kolhe[2,4], Susan Sanchez[1,2], Yung-Yi C. Mosley[1,3], Ralph A. Tripp[1,2]\***

**1** Department of Infectious Disease, College of Veterinary Medicine, University of Georgia, Athens, Georgia, United States of America, **2** State of Georgia COVID-19 Taskforce, Athens, Georgia, United States of America, **3** Tifton Diagnostic and Investigational Laboratory, University of Georgia, Athens, Georgia, United States of America, **4** Medical College of Georgia, Augusta University, Augusta, Georgia, United States of America

\* ratripp@uga.edu

## Abstract

The COVID-19 pandemic caused by the SARS-CoV-2 is a serious health threat causing worldwide morbidity and mortality. Real-time reverse transcription PCR (RT-qPCR) is currently the standard for SARS-CoV-2 detection. Although various nucleic acid-based assays have been developed to aid the detection of SARS-CoV-2 from COVID-19 patient samples, the objective of this study was to develop a diagnostic test that can be completed in 30 minutes without having to isolate RNA from the samples. Here, we present an RNA amplification detection method performed using reverse transcription loop-mediated isothermal amplification (RT-LAMP) reactions to achieve specific, rapid (30 min), and sensitive (<100 copies) fluorescent detection in real-time of SARS-CoV-2 directly from patient nasopharyngeal swab (NP) samples. When compared to RT-qPCR, positive NP swab samples assayed by fluorescent RT-LAMP had 98% (n = 41/42) concordance and negative NP swab samples assayed by fluorescent RT-LAMP had 87% (n = 59/68) concordance for the same samples. Importantly, the fluorescent RT-LAMP results were obtained without purification of RNA from the NP swab samples in contrast to RT-qPCR. We also show that the fluorescent RT-LAMP assay can specifically detect live virus directly from cultures of both SARS-CoV-2 wild type (WA1/2020), and a SARS-CoV-2 B.1.1.7 (alpha) variant strain with equal sensitivity to RT-qPCR. RT-LAMP has several advantages over RT-qPCR including isothermal amplification, speed (<30 min), reduced costs, and similar sensitivity and specificity.

## Introduction

COVID-19 is a pandemic disease caused by SARS-CoV-2 [1, 2]. The animal reservoir for SARS-CoV-2 is unknown, but it is believed to be of zoonotic origin as SARS-CoV-2 isolates from bats and pangolins are genetically related to human isolates [3, 4]. SARS-CoV-2 has initiated a pandemic, and according to the WHO Coronavirus (COVID-19) Detailed Surveillance Data Dashboard, currently there are more than 170 million confirmed COVID-19 cases and

**Data Availability Statement:** All relevant data are within the manuscript and its Supporting Information files.

**Funding:** Funding was through the Georgia COVID-19 Task Force and the Georgia Research Alliance to RT. The funders had no role in study design, data collection and analysis, decision to publish, or preparation of the manuscript.

**Competing interests:** he authors have declared that no competing interests exist.

more than 3.8 million deaths from COVID-19 worldwide. SARS-CoV-2 is an RNA virus with a genome that encodes four structural proteins, i.e. the spike (S) protein, the envelope (E) protein, the membrane (M) protein and the nucleocapsid (N) protein, and the remainder polyproteins that are cleaved into various non-structural proteins important for replication [5, 6]. SARS-CoV-2 infects a variety of cell types by binding to its receptor, i.e. angiotensin-converting enzyme-2 (ACE2) by the receptor-binding domain (RBD) on the spike (S) protein that is proteolytically-activated by human serine proteases [7, 8]. Cell entry depends on binding of the surface unit, S1, of the S protein to ACE2 to facilitate viral attachment to the target cells, and entry requires S protein cleavage at the S1/S2 and the S2' site and allows fusion of viral and cellular membranes, a process driven by the S2 subunit [9]. The virus is transmitted from person to person through aerosol [10, 11]. The pathological symptoms of infection can vary but include influenza-like respiratory illness, dry cough, fever, and pulmonary complications [12], however many SARS-CoV-2 infected individuals are asymptomatic and capable of increasing the spread of the disease [13, 14]. Researchers are currently testing 100 vaccines in clinical trials on humans, and several have reached the final stages of testing. Three COVID-19 vaccines to date have been authorized and recommended for use in the United States, i.e. the Pfizer-BioNTech mRNA vaccine, the Moderna mRNA vaccine, and the Janssen-Johnson & Johnson viral vector vaccine [15]. Therefore, precautions to control disease and infection prevention practices are critical and need to be supported by diagnostics, which is a crucial component when caring for a patient with suspected or confirmed SARS-CoV-2 infection.

In addition to the Centers for Disease Control and Prevention (CDC) RT-PCR SARS-CoV-2 test [16], many other diagnostic tests have recently become available. RT-qPCR is the gold standard in diagnostic testing but its use is expensive, time-consuming, and labor-intensive. An advantage of RT-qPCR is the ability to tune the reaction using multiple temperatures allowing for multiplex target discrimination, optimized primer annealing, denaturation, and extension. Further, these properties can be adjusted at each cycle, and RT-qPCR can monitor reaction progress by non-specific dyes (e.g., SYBR Green) or TaqMan probes [17]. However, isothermal detection platforms offer key advantages over RT-qPCR platforms in terms of reduced cost, speed, and reduced labor yet have the same sensitivity. Specifically, isothermal detection platforms allow amplification at constant temperatures and avoid the need for thermal cycling required for RT-qPCR. Here, we describe a one-step isothermal amplification/detection method that is amenable to lab-based or POC/field deployment. Loop-mediated isothermal amplification (LAMP) reactions use a strand-displacing DNA polymerase (and reverse transcriptase for RNA targets) with four to six primers resulting in exponential amplification [18, 19]. LAMP is a one-step reaction allowing all the primers and enzymes to be incubated isothermally in a single step. Reverse transcription LAMP (RT-LAMP) is useful for detecting RNA-based viruses as it combines conventional LAMP with a reverse transcriptase enzyme, allowing for simultaneous reverse transcription and amplification. The endpoint for LAMP can be real-time detection using fluorescence or absorbance [20]. The RT-LAMP assay has been shown to detect SARS-CoV-2 in ≤30 minutes using patient samples such as saliva and nasopharyngeal swabs [21, 22]. However, to achieve these results, the sample preparation includes a viral RNA extraction step that increases the cost and time required to perform the assay. The LAMP detection step is performed using direct visual assessment of color change using an indicator to show a positive reaction, e.g. a shift from dark blue to lighter blue using hydroxynaphthol blue [23, 24], or pink to yellow using creosol red [25]. However, the assessment of color change has issues including unexpected signals derived from primer-dimer and/or non-primer reactions. Fluorescent dyes can be used in the LAMP reaction for fluorescence monitoring which improves detection and solves the problem of non-primer-derived signals [26, 27]. Thus, RT-LAMP has several advantages over RT-qPCR, and when fluorescence is

used as an endpoint for RT-LAMP there is also improved detection arguing fluorescent RT-LAMP superiority [28]. Further, fluorescent RT-LAMP can be deployed as a point-of-care (POC) test filling a role that is vital for disease management and control.

# Materials and methods

## Fluorescent RT-LAMP reactions

All RT-LAMP reactions were performed using a Warm Start LAMP 2X master mix (New England Biolabs, Ipswitch, MA, USA) and an AriaMX Real-Time PCR System machine (Agilent Technologies, Santa Clara, CA) for 30 min and monitored once per minute for fluorescence in the SYBR/FAM channel. Reactions contained warm Start LAMP Kit 2X Master Mix (12.5 µl NEB), LAMP primer mix (10X, 2.5 µl, IDT), assay target (RNA, or live virus, or clinical specimen, 2 µl), SYBR-like fluorescent dye diluted 8-fold in deionized water (50X, 0.5 µl, NEB) and deionized water (7.5 µl) for a total of 25 µl. The 10X LAMP primer mix contained FIP/BIP at 10µM, LF/ LB at 2 µM and 4 µM, respectively, prepared in deionized RNAse DNAse free water. Primers were designed from multiple alignments of SARS-CoV-2 N gene (nts 251–468) with (USA-WA1/2020, GenBank accession number MN985325.1) as a reference genome using PrimerExplorer V software. Primers were commercially synthesized by Integrated DNA technologies (IDT, Coralville, IA) (S1 Table). Final concentrations of RT-LAMP primers LAMP primers (S1 Table) were used for the fluorescent reaction: FIP/BIP 1.0 µM, F3/B3, and LF/LB primers were used at 0.2 µM and 0.4 µM, respectively.

## Preparation of RT-LAMP templates

**NP swab samples / cell culture derived virus.**   For NP swabs, a 20 µl aliquot was mixed 1:1 (v:v) with 2X LAMP Lysis Buffer (LLB): 2% Triton X-100 (Sigma, St. Louis, MO) adjusted to pH = 8.0 with 2M Tris-HCL (Sigma, St. Louis, MO) and 80 U/ml Proteinase K (NEB) followed by incubation at room temperature (RT) for 15 min, then incubation at 95˚C for 10 min to inactivate the Proteinase K. The patient sample lysate (2 µl) was used as target directly in the fluorescent RT-LAMP reactions without further processing. Fluorescent RT-LAMP reactions using virus-infected cell culture supernatant samples were used to detect live viruses. SARS-CoV-2 (USA-WA1/2020, GenBank: MN985325.1), and alpha variant B1.1.7 (20I/501Y.V1, GenBank: MW422255.1), human coronavirus strains, OC43 (GenBank: AY585228.1), 229E (GenBank: AF304460.1), and NL63 (GenBank: AY567487.2) (BEI Resources, USA) were used to infect Vero cells and cultured to $10^7$ PFU/ml. A 20 µl aliquot of the infected cell culture supernatants was mixed with 20 µl of 2X LLB and incubated at RT for 15 min followed by 10 min 95˚C incubation to inactivate Proteinase K. Fluorescent RT-LAMP was performed using 2 µl of the sample lysate.

**Fluorescent RT-LAMP reaction using virus-infected cell culture supernatant samples.** An identical process was used to detect live viruses. SARS-CoV-2 (USA-WA1/2020, GenBank: MN985325.1), and alpha variant B.1.1.7 (20I/501Y.V1, GenBank: MW422255.1) were inoculated on Vero cells and cultured to $10^7$ pfu/ml. A 20 µl aliquot of the infected cell culture supernatants was mixed with 20 µl of 2X LLB and incubated at RT for 15 min followed by 10 min 95˚C incubation to inactivate Proteinase K. Fluorescent RT-LAMP was performed using 2 µl of the sample lysate. We also examined human coronavirus strains, OC43 (GenBank: AY585228.1), 229E (GenBank: AF304460.1), and NL63 (GenBank: AY567487.2), all of which were obtained from BEI Resources and cultured using Vero cells for detection in fluorescent RT-LAMP as specificity controls.

## Optimizing the limit of detection/fluorescent end-point cutoff

The fluorescent LAMP reaction contained an intercalating fluorescent SYBR-like dye with excitation ($\lambda_{Exci}$ = 497 nm) and emission ($\lambda_{Emis}$ = 520 nm) when complexed with the double-stranded DNA LAMP reaction product. This allowed for easy monitoring of the accumulation of double-stranded DNA in the RT-LAMP reaction with a real-time thermocycler using a standard SYBR/FAM channel for fluorescence. Using a log dilution series (ranging from $10^8$ to 10 copies of the N gene) of purified RNA templates described below (N gene IVT RNA and viral genomic RNA), we measured the signal from the fluorescent RT-LAMP reaction at 1 min intervals analogous to a standard dye-based RT-qPCR reaction utilizing a 30 min total time cut-off. The results were compared to the cycle threshold (Ct) values obtained from our RT-qPCR assay (described below) for the same purified RNA templates. In the fluorescent RT-LAMP reaction, the purified RNA templates representing ~100 copies or more of the N gene generated strong fluorescence by 30 min and were readily detectable by RT-qPCR. RNA templates diluted beyond 10 copies of the N gene did not generate fluorescence signal within 30 min in fluorescent RT-LAMP and were negative in RT-qPCR with a cut-off of 40 cycles. Based on this observation, and consistent with other related RT-LAMP publications [29, 30], we used fluorescence at 30 min to determine whether a sample was positive or negative.

## RT-qPCR

Serial log dilutions of a 2019-nCoV_N_positive control plasmid (IDT) containing the SARS-CoV-2 (Wuhan-Hu-1 strain, GenBank: MN908947) N gene from $10^5$ copies/μl to 1 copy/μl, or SARS-CoV-2 genomic viral RNA (vRNA), or SARS-CoV-2 N gene *in vitro* transcribed RNA (IVT RNA) from 1ng/μl to 0.1fg/μl corresponding to $10^7$ copies/μl to 1 copies/μl, or $10^8$ copies/μl to 10 copies/μl, respectively, were prepared in nuclease-free water. Nuclease-free water was used as a no-template control for all reactions. The SARS-CoV-2 RNA templates were added to LunaScript RT SuperMix (NEB) at 2 μl in a 20 μl reaction and reverse transcribed for 10 min at 55°C. The cDNA from this reaction was used directly at 1 μl in a 20 μl reaction with Luna Universal Probe qPCR Master Mix (NEB) and 1 μl of the SARS-CoV-2 CDC EUA N1 probe assay (IDT) and amplified for 40 cycles in a two-step reaction (15s/95°C, and 30s/60°C) after an initial 1 min pre-amplification hold at 95°C on an AriaMx Real-Time PCR System (Agilent). SARS-CoV-2 plasmid DNA was RT-qPCR amplified by adding 1 μl to a 20 μl reaction with the same qPCR master mix and probe assay using the same two-step amplification protocol after an initial 3 min pre-amplification hold at 95°C. Results from the N gene control plasmid were used to construct a standard curve relating the N gene copy number to RT-qPCR Ct values (S3 Fig). The N gene copy numbers of the SARS-CoV-2 IVT RNA or SARS-CoV-2 genomic vRNA per unit mass were extrapolated from this curve.

For analysis of clinical samples, RNA was extracted from 200 μl of patient nasopharyngeal swab liquid in saline using the RNAdvance Viral Reagent Kit (Beckman Coulter, Indianapolis, IN) and eluted in a final volume of 40 μl of nuclease-free water. RT-qPCR analysis of purified RNA from patient samples as described above was performed using the CDC EUA N1 probe assay and the CDC EUA human RNAseP internal control probe assay to validate sample RNA extraction in separate reactions.

## SARS-CoV-2 RNA

Genomic RNA from SARS-CoV-2 (USA-WA1/2020 GenBank: MN985325.1) and the SARS-CoV-2 variant B.1.1.7 (SARS-CoV-2/human/USA/SEARCH-5574-SAN/2020 GenBank: MW422255.1) was purified from 200 μl of virus-infected Vero cell culture supernatant using

an RNAeasy kit (Qiagen) according to the manufacturer's instructions and eluted in 40 μl of nuclease-free deionized water and stored at -80˚C.

### *In vitro* transcribed (IVT) RNA

The 1260 bp complete coding sequence of SARS-CoV-2 Wuhan-Hu-1 strain N gene (Gen-Bank: MN908947) was amplified from 2019-nCoV_N_Positive Control plasmid (IDT) using primers (IDT) SARS2N_F and SARS2N_R (S1 Table) in standard end-point PCR with Q5 DNA polymerase (NEB). The PCR band representing the N gene coding sequence was sub-cloned into plasmid pMiniT2.0 (NEB) with the 5' end of the coding sequence proximal to the vector-provided T7 polymerase promoter and confirmed by Sanger sequencing. HiScribe T7 RNA Synthesis Kit (NEB) was used to produce the SARS-CoV-2 N gene single-stranded positive-sense RNA via in-vitro transcription from the pMiniT2.0 construct. The IVT RNA was digested with DNase1 (NEB) and purified using RNAClean XP (Beckman Coulter). Purified IVT RNA was eluted in nuclease-free water, adjusted to a final concentration of 1 μg/μl in nuclease-free water, and stored at -80˚C.

### Analysis of NP swab samples

Deidentified samples were collected from consenting adult volunteers by the Georgia National Guard or the University of Georgia (UGA) COVID-19 Task Force following written approval and were tested from frozen -80˚C samples collected by the Georgia Taskforce for COVID-19, the Georgia National Guard, or the University of Georgia Diagnostic Laboratories. NP samples were collected using a sterile swab applicator that was placed in 1 mL of saline. The University of Georgia (UGA) Diagnostic Laboratories determined the SARS-CoV-2 status of the NP samples using an Applied Biosystems TaqPath COVID-19 EUA assay (ThermoFisher) to detect SARS-CoV-2 RNA in a multiplex RT-qPCR format. The assay is specific for three SARS-CoV-2 genome regions, i.e. spike (S) gene, nucleocapsid (N) gene, and orf-1ab. We analyzed the same NP samples using the CDC EUA N1 RT-qPCR assay for the detection of SARS-CoV-2 RNA. The CDC EUA N1 RT-qPCR assay is singleplex and specific for the SARS-CoV-2 N gene. Fifty SARS-CoV-2 positive samples and 84 SARS-CoV-2 negative samples were tested in our laboratory using the CDC EUA N1 RT-qPCR assay to detect the N gene sequence. All NP samples were subsequently tested in a fluorescent RT-LAMP assay for comparative SARS-CoV-2 diagnostic analysis with CDC EUA N1 RT-qPCR assay.

## Results

### Analytical sensitivity of fluorescent RT-LAMP using IVT RNA and genomic viral RNA

We examined the sensitivity of the fluorescent RT-LAMP assay, by comparison, to the CDC EUA N1 RT-qPCR assay. The limit of detection (LOD) in the RT-qPCR assay using SARS-CoV-2 (Wuhan-Hu-1) IVT RNA was 0.1 fg of RNA corresponding to 10 copies of the N gene sequence (Fig 1). The LOD by RT-qPCR was 100 copies of purified viral genomic RNA from SARS-CoV-2 USA-WA1/2020 (S1A Fig). The fluorescent RT-LAMP assay rapidly and sensitively detected SARS-CoV-2 IVT RNA with a LOD of approximately 10–100 N gene copies with efficient isothermal amplification across seven logs of target concentration (Fig 2). The fluorescent RT-LAMP assay can detect 100 copies of SARS-CoV-2 (USA-WA1/2020 GenBank: MN985325.1) viral genomic RNA equaling the sensitivity of RT-qPCR (S1B Fig).

**A.** SARS-CoV-2 N Gene IVT RNA Amplification Curve

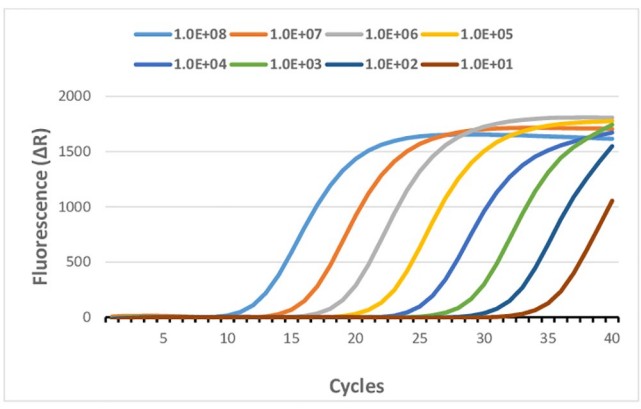

**B.** Schematic of the pMiniT2.0 SARS-CoV-2 Wuhan-Hu-1 N gene in-vitro transcription construct

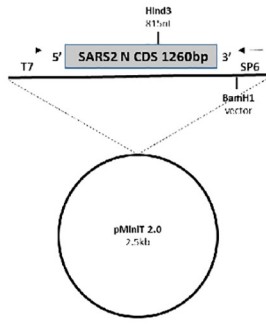

**C.** Purified SARS-CoV-2 Wuhan-Hu-1 N gene IVT RNA



**Fig 1. SARS-CoV-2 RT-qPCR CDC EUA N1 probe assay. A.** RT-qPCR (CDC EUA N1 probe assay) targeting SARS-CoV-2 Wuhan-Hu-1 N gene *in vitro* transcribed (IVT) RNA. Log dilution series of N gene IVT RNA ranging from $10^8$ to 10 copies of the N gene underwent reverse transcription. Fluorescence data were collected and processed using a 40 cycle cut-off on an AriaMx Real-Time PCR System. **B.** Schematic of the pMiniT2.0 SARS-CoV-2 Wuhan-Hu-1 N gene *in vitro* transcription construct with T7 RNA polymerase promoter driving full-length (1260 nt) N gene RNA production. **C.** DNase-treated and purified SARS-CoV-2 Wuhan-Hu-1 N gene IVT RNA on a non-denaturing 1% agarose gel stained with ethidium bromide.

**A.** Amplification Curve Fluorescent RT-LAMP

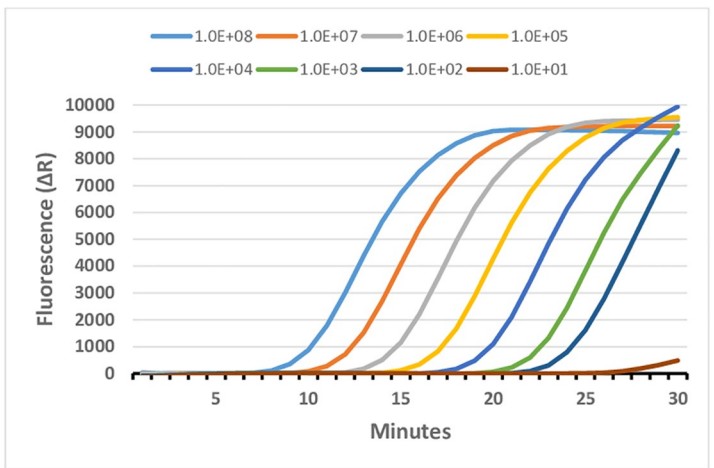

**B.** Standard Curve Fluorescent RT-LAMP assay

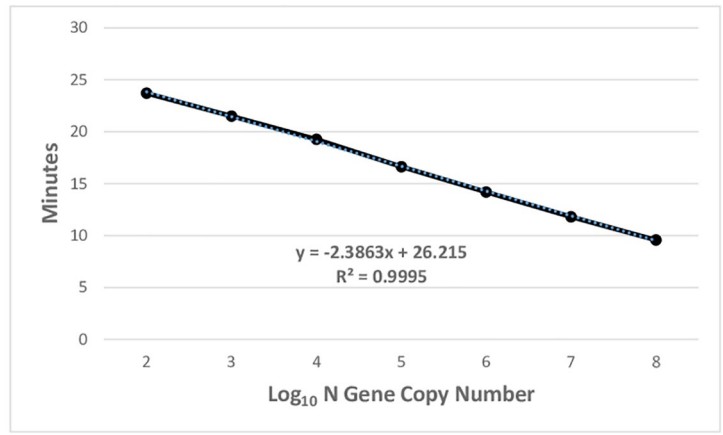

**C.** Melt Curve Fluorescent RT-LAMP assay

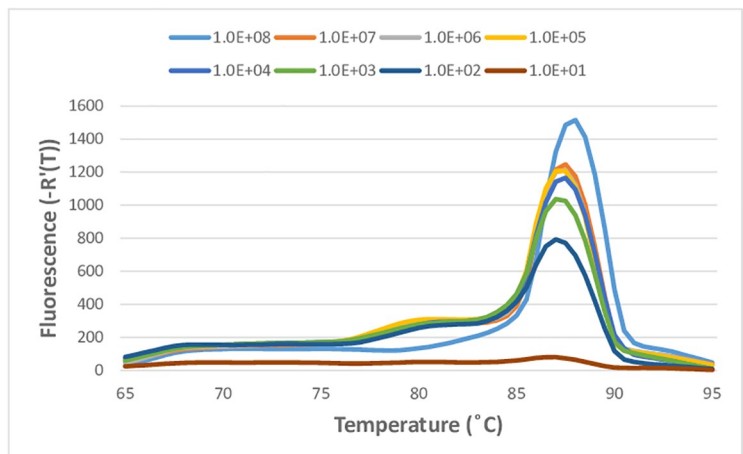

**Fig 2. SARS-CoV-2 fluorescent RT-LAMP assay. A.** Fluorescent RT-LAMP assay targeting SARS-CoV-2 Wuhan-Hu-1 N gene IVT (GenBank: MN908947) RNA. Log-dilution series of N gene IVT RNA ranging from $10^8$ to 10 copies

of the N gene was added to fluorescent RT-LAMP reactions containing six LAMP primers (FIP/BIP, F3/B3, and LF/LB) and an SYBR-like fluorescent dye. Reactions were performed on an AriaMx Real-Time PCR System at 65˚C for 30 min with fluorescence monitored in the FAM/SYBR channel. Fluorescence values were plotted against the elapsed reaction time (min) to generate the amplification curve. **B**. Standard Curve of fluorescent RT-LAMP assay **C**. Melt curve of the fluorescent RT-LAMP products produced from the reactions in Fig 2A.

## Fluorescent RT-LAMP detects the SARS-CoV-2 B.1.1.7 (alpha) variant

SARS-CoV-2 (WA1/2020) or SARS-CoV-2 B1.1.7 (alpha) variant (20I/501Y.V1) infected (MOI = 0.1) Vero cell culture supernatants were examined by fluorescent RT-LAMP. We demonstrated that the fluorescent RT-LAMP assay successfully detected the SARS-CoV-2 B1.1.7 (alpha) variant with equal sensitivity to that of SARS-CoV-2 at a LOD of approximately 100 pfu/ml (Fig 3A and 3B). To confirm RT-LAMP specificity, we tested human coronavirus strains, OC43 (GenBank: AY585228.1), 229E (GenBank: AF304460.1), and NL63 (GenBank: AY567487.2) infected (MOI = 1.0) Vero cells. All human coronavirus strains tested did not produce a detectable fluorescent signal in the fluorescent RT-LAMP assay (Fig 3C). The sensitivity of fluorescent RT-LAMP detection of both the variant and the wild-type SARS-CoV-2 strains compared favorably with the RT-qPCR results for RNA purified from an equivalent amount of SARS-CoV-2 infected Vero cell culture supernatant (Fig 4).

## Fluorescent RT-LAMP detects SARS-CoV-2 in nasopharyngeal swab samples

We determined the presence of SARS-CoV-2 directly in clinical nasopharyngeal swab samples collected from patients using fluorescent RT-LAMP. Nasal swabs and NP swabs from suspected COVID-19 patients were collected by personnel from the Georgia COVID-19 Task Force and the University of Georgia Diagnostic Laboratories and confirmed positive or negative for SARS-CoV-2 using an Applied Biosystems TaqPath COVID-19 EUA kit to detect SARS-CoV-2 RNA in a multiplex RT-qPCR format. In comparing fluorescent RT-LAMP to RT-qPCR, we chose to evaluate the singleplex N gene-specific SARS-CoV-2 CDC EUA N1 RT-qPCR assay since it shares close target specificity with our fluorescent RT-LAMP assay (S1 Table). Using the same NP samples, we used the CDC EUA N1 RT-qPCR assay on both SARS-CoV-2 positive (n = 50) and SARS-CoV-2 negative (n = 84) NP samples in triplicate. Positive samples in the CDC EUA N1 RT-qPCR probe assay with Ct values <40 for each of the three repeat samples were considered N gene positives in our comparative analysis. We determined that 84% (n = 42/50) of the previously established SARS-CoV-2 positive NP samples were N gene-positive (Ct<40) by the CDC EUA N1 RT-qPCR assay for each of three repeat measurements (Table 1). The mean Ct of the SARS-CoV-2 N gene-positive NP samples (n = 42) ranged from 16 to 34 cycles with an overall mean Ct of 25.7, 95% CI [24–27] using the SARS-CoV-2 CDC EUA N1 RT-qPCR assay. Negative samples with no Ct value in the CDC EUA N1 RT-qPCR assay for any of three repeat measurements were considered true N gene negatives. For the previously confirmed SARS-CoV-2 negative NP samples, we determined that 81% (n = 68/84) gave matching N gene negative results, i.e. no Ct value in any of the three repeat measurements using the SARS-CoV-2 CDC EUA N1 RT-qPCR assay (Table 2).

The fluorescent RT-LAMP assays were performed in triplicate using the same NP samples that were previously determined to be positive for SARS-CoV-2 (n = 50) by the UGA Diagnostic Laboratories. The fluorescent RT-LAMP assay values for the SARS-CoV-2 N gene-positive NP samples ranged from 18 to 28 minutes with an overall mean of 23 minutes and 95% CI [22–24]. The results for N gene-positive NP samples showed 98% (n = 41/42) concordance with RT-qPCR for SARS-CoV-2 positivity (Table 1). N gene NP samples with no Ct result in

**A.** SARS-CoV-2 WA1/2020 Cell Culture Lysate

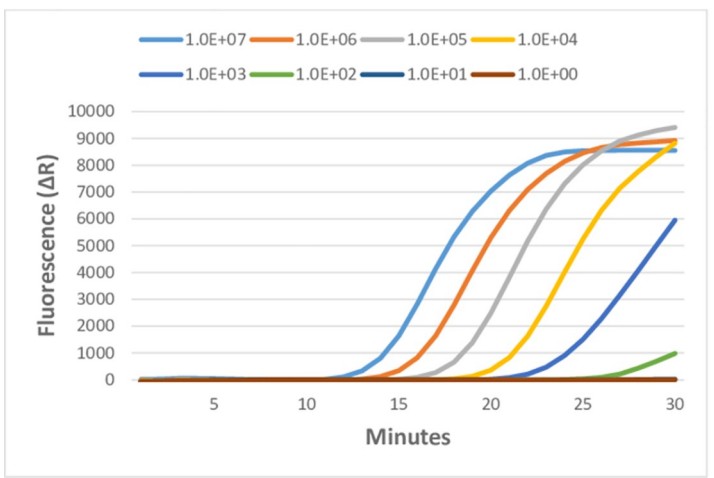

**B.** SARS-CoV-2 B.1.1.7 (alpha) variant (20I/501Y.V1) Cell Culture Lysate

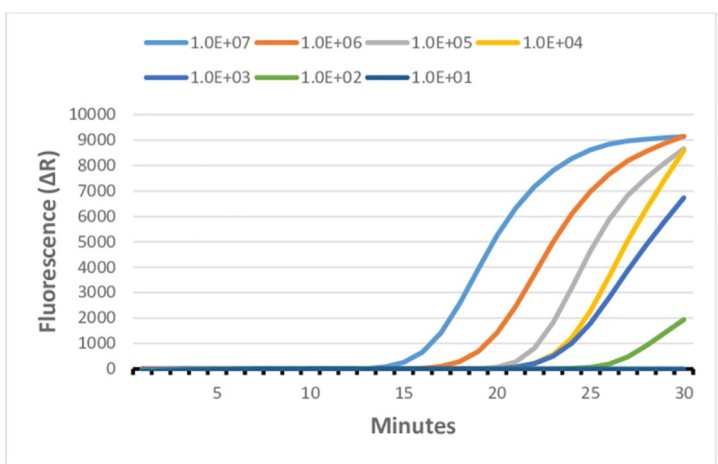

**C.** Negative Control Cell Culture Lysates

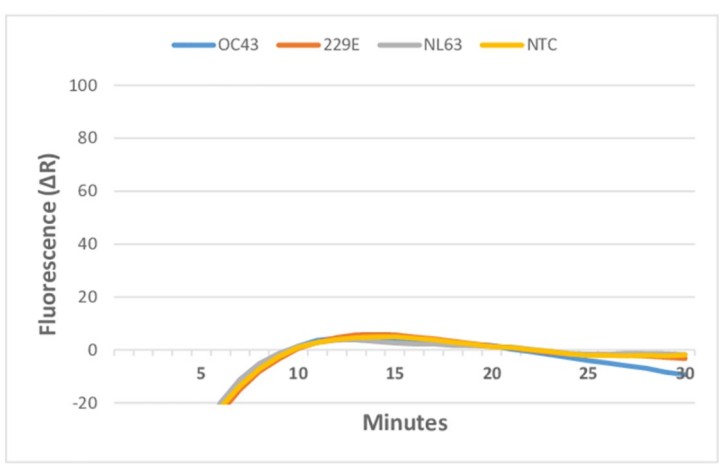

**Fig 3. SARS-CoV-2 live virus fluorescent RT-LAMP assay. A.** Fluorescent RT-LAMP assay targeting SARS-CoV-2 (USA-WA1/2020 GenBank: MN985325.1), and **B**. Fluorescent RT-LAMP assay targeting SARS-CoV-2 variant B1.1.7 variant (20I/501Y.V1) in infected Vero cell culture. Culture supernatants were log-serially diluted from $10^7$ pfu/ml to 10 pfu/ml. Fluorescent RT-LAMP was performed using six LAMP primers (FIP/BIP, F3/B3, and LF/LB), and SYBR-like fluorescent dye. Reactions were performed on an AriaMx Real-Time PCR System at 65˚C for 30 min with fluorescence monitored in the FAM/SYBR channel. Fluorescence values were plotted against the elapsed reaction time (min) to generate the amplification curves. Virus-infected Vero cell culture supernatants below $10^2$ pfu/ml were not detected by 30 min. **C.** No Template Control (NTC), and heterologous human coronavirus strains at $10^6$ pfu/ml, OC43 (GenBank: AY585228.1), 229E (GenBank: AF304460.1), and NL63 (GenBank: AY567487.2) in infected Vero cells failed to produce fluorescence signal over the background in 30 min in the fluorescence RT-LAMP assay.

### A. SARS-CoV-2 WA1/2020 Cell Culture Lysate

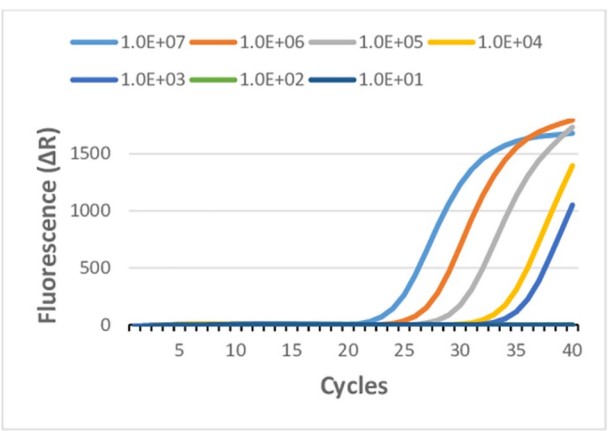

### B. SARS-CoV-2 B.1.1.7 (alpha) variant (20I/501Y.V1) Cell Culture Lysate

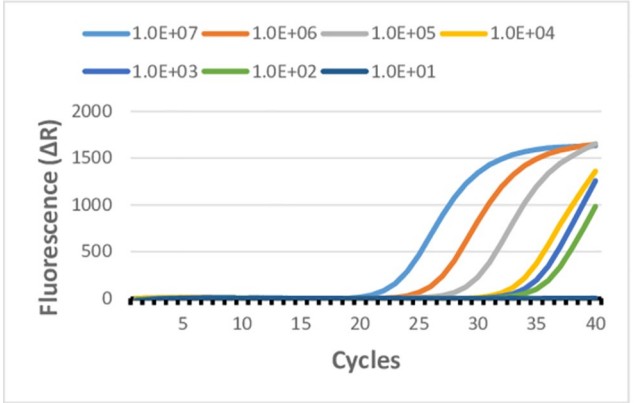

**Fig 4. SARS-CoV-2 live virus CDC EUA N1 RT-qPCR assay. A.** RT-qPCR using CDC EUA N1 probe assay targeting SARS-CoV-2 (WA1/2020) and, **B.** targeting SARS-CoV-2 B.1.1.7 (Alpha) (20I/501Y.V1) in infected Vero cells. Culture supernatants were log-serially ranging from $10^7$ pfu/ml to 10 pfu/ml. RNA was purified from cell culture supernatants and reverse transcribed and cDNA was used in an RT-qPCR reaction in the CDC EUA N1 Probe Assay. Fluorescence data were collected and processed using a 40 cycle cut-off on an AriaMx Real-Time PCR System.

**Table 1. Fluorescent RT- LAMP on SARS-CoV-2 positive NP swab samples.**

| SARS-CoV-2 CDC EUA N1 RT-qPCR Mean Ct [95%CI] | Number of Samples | RT-LAMP Concordant Results (%) | RT-LAMP Discordant Results (%) | Mean RT-LAMP (min) [95% CI] |
|---|---|---|---|---|
| **Below assay cut-off (Ct<40)** M = 25.7, 95% CI [24.0–27.3] | 42 | 41 (98) | 1 (2) | mean = 23 |
| | | | | 95% CI [22– 24] |
| **No Ct** | 8 | 1 (12) | 7 (88) | na |
| **Positive samples** | 50 | 42 (84) | 8 (16) | na |

Patient samples were previously confirmed as SARS-CoV-2 positive or negative by the UGA Diagnostic Laboratories using the ThermoFisher Applied Biosystems TaqPath COVID-19 EUA Combo *Kit* to detect SARS-CoV-2 RNA in a multiplex RT-qPCR format specific for three SARS-CoV-2 genome regions: S gene, N gene, and orf-1ab. We further analyzed the same NP samples by RT-qPCR using the singleplex CDC EUA N1 probe assay with a cut-off of 40 cycles, and the N gene-specific fluorescent RT-LAMP assay with a cut-off at 30 min. FIP/BIP, F3/B3, and LF/LB primers and an SYBR-like fluorescent dye were used in the RT-LAMP assay. The SARS-CoV-2 positive sample fluorescent RT-LAMP results (n = 50) were grouped according to their CDC EUA N1 RT-qPCR assay result: samples below the assay cut-off (Ct<40) n = 42/50, or samples above the assay cut-off (no Ct) n = 8/50. The number of positive samples having a fluorescent RT-LAMP assay value less than the 30 min assay cut-off (n = 41/42) has concordant results to RT-qPCR for these samples. One specimen (n = 1/42) had discordant results in the fluorescent RT-LAMP assay as it failed to produce a signal above the background in the 30 min assay time. The average fluorescent RT-LAMP assay value (23 minutes) is calculated for the group of positive samples (n = 41) at the 95%CI, 23 [22–24]. Samples having (no Ct) result in RT-qPCR (8/50) had discordant fluorescent RT-LAMP results signified by a positive signal in the RT-LAMP assay (n = 7/8). Fluorescent LAMP is measured by the time in minutes required to generate fluorescence signal (ΔR) over baseline in the assay.

the CDC EUA N1 RT-qPCR assay (8/50) had discordant fluorescent RT-LAMP results as indicated by a positive signal in the RT-LAMP assay (n = 7/8).

A fluorescent RT-LAMP assay was performed on the same NP samples that were previously shown to be SARS-CoV-2 negative (n = 84) by the UGA Diagnostic Laboratories. Results for the fluorescent RT-LAMP assay on the N gene negative NP samples showed 87% (n = 59/68) concordance with RT-qPCR for SARS-CoV-2 (Table 2), while 13% (n = 9/68) of N gene negative samples gave discordant fluorescent RT-LAMP assay results as determined by a positive result in the 30 minute assay time. When compared to RT-qPCR, N gene-positive samples assayed by fluorescent RT-LAMP had 98% (n = 41/42) concordance, while fluorescent RT-LAMP values were 87% (n = 59/68) concordant with RT-qPCR. Importantly, the

**Table 2. Fluorescent RT- LAMP of SARS-CoV-2 negative NP swab samples.**

| SARS-CoV-2 CDC EUA N1 RT-qPCR Result | Number of Samples | RT-LAMP Concordant Results (%) | RT-LAMP Discordant Results (%) |
|---|---|---|---|
| **No Ct** | 68 | 59 (87) | 9 (13) |
| **Below assay cut-off (Ct<40)** | 16 | 7 (44) | 9 (56) |
| **All Negative samples** | 84 | 66 (79) | 18 (21) |

NP samples were previously confirmed as SARS-CoV-2 positive or negative by the UGA Diagnostic Laboratories using the ThermoFisher Applied Biosystems TaqPath COVID-19 EUA *Kit* to detect SARS-CoV-2 RNA in a multiplex RT-qPCR format specific for three SARS-CoV-2 genome regions: S gene, N gene, and orf-1ab. We further analyzed the same NP samples by RT-qPCR using the singleplex CDC EUA N1 probe assay with a cut-off of 40 cycles, and the N gene-specific fluorescent RT-LAMP assay with a cut-off at 30 min. FIP/BIP, F3/B3, and LF/LB primers and an SYBR-like fluorescent dye were used in the RT-LAMP assay. The SARS-CoV-2 negative sample fluorescent RT-LAMP results (n = 84) were grouped according to their CDC EUA N1 RT-qPCR assay result: samples showing No Ct (n = 68/84), or samples having Ct below the assay cut-off Ct<40 (n = 16/84). N = 68 of the negative NP samples failed to signal in our RT-qPCR assay, as expected. Of this group, n = 59/68 gave concordant fluorescent RT-LAMP assay results failing to signal above baseline in the 30 min assay time, and n = 9/68 gave discordant fluorescent RT-LAMP assay results signified by a positive fluorescent RT-LAMP signal in the 30 min assay time. Some of the negative NP samples gave RT-qPCR Ct values below the assay cut-off (n = 16/84) suggesting a positive result. This is unexpected from negative samples but likely is related to the differential specificities between our singleplex CDC EUA N1 RT-qPCR assay and the multiplex RT-qPCR assay used by the UGA Diagnostic Laboratories. Of this group of samples, n = 7/16 also gave a positive fluorescent RT-LAMP signal, while n = 9/16 failed to signal above baseline in the 30 min assay time. Fluorescent RT-LAMP is measured by the time in minutes required to generate fluorescence signal (ΔR) over baseline in the assay.

fluorescent RT-LAMP assay results were obtained without purification of RNA from the NP samples in contrast to the RT-qPCR assay protocol.

## Discussion

The benchmark method for the detection of SARS-CoV-2 from patient samples is the RT-qPCR assay requiring RNA extraction, highly trained laboratory personnel, and expensive equipment and reagents. Currently, the majority of clinical diagnosis for SARS-CoV-2 are performed by central testing laboratories that typically takes >24h for return of results. These clinical RT-qPCR assays are used typically with symptomatic patients, are not required to be low-cost, but rely on their analytical sensitivity to produce a definitive diagnosis even if only a single testing opportunity is available. Diagnostic assays used in surveillance are designed to detect the prevalence of SARS-CoV-2. These tests should be cost-effective, rapid, and easy to perform allowing for frequent testing. We sought to improve upon SARS-CoV-2 diagnostic testing by reducing the time for confirmation of a result, lowering costs to acquire the results, and developing a reliable, sensitive, and specific rapid assay. Many emerging studies have shown that RT-LAMP can meet these requirements for screening and testing for SARS-CoV-2 and potentially be a complementary tool to RT-qPCR [18, 31, 32]. Several RT-LAMP studies have focused on developing point-of-care (POC) diagnostic assays by detection of SARS-CoV-2 RNA in clinical samples [33, 34]. Other POC RT-LAMP studies have developed assays to allow multiplexing of more than one genomic target, while others have focused on optimizing RT-LAMP conditions to improve detection of SARS-CoV-2 genomes [35], but direct detection of the virus has been limited.

In this study, we show that fluorescent RT-LAMP has similar performance to RT-qPCR in detecting SARS-CoV-2 from the same clinical samples, and importantly, that fluorescent RT-LAMP can sensitively and reliably detect virus from only minimally pre-processed nasal swabs. Despite differences in SARS-CoV-2 detection in NP samples between the multiplex assay and the RT-LAMP assay, we believe that comparative sensitivity and specificity is important for accelerating diagnostic applications. Likely, the broad target specificity associated with the multiplex RT-qPCR assay used by the UGA Diagnostic Laboratories may in part account for the difference in sensitivity we obtained on the same NP samples using the singleplex CDC EUA N1 RT-qPCR assay. It is also possible that RNA degradation of the NP samples may have occurred by the time we obtained them as they were stored in saline and subsequently underwent several freeze-thaw cycles which may have contributed to differences in sensitivity associated with our RT-qPCR assay.

Of the group of SARS-CoV-2 positive NP samples from the UGA Diagnostic Laboratories, we found 84% (n = 42/50) to be N gene-positive as determined by our RT-qPCR assay. Importantly, fluorescent RT-LAMP assay results were 98% (n = 41/42) concordant as N gene-positive samples. Eight SARS-CoV-2 positive NP samples determined by the UGA Diagnostic Laboratories failed to amplify the N gene target in our RT-qPCR assay. However, the internal control amplicon (human RNase P) amplified in all NP samples tested s that the extracted RNA was intact. We found n = 7/8 positive NP samples by the fluorescent RT-LAMP assay for the failed RT-qPCR NP samples which was discordant with our RT-qPCR result for these NP samples but in agreement with NP samples showing SARS-CoV-2 positive standing by the UGA Diagnostic Laboratories. This may suggest that sample preparation differences (e.g. RNA purification for RT-qPCR or the minimal pre-processing for RT-LAMP) may contribute to the different results. We show that the fluorescent RT-LAMP assay sensitivity is more than sufficient to reliably detect SARS-CoV-2 in NP samples to virus load, as SARS-CoV-2 patients have been reported to have a range from 641 copies/mL to $10^{11}$ copies/mL with a median of

$10^4$–$10^5$ [36]. Of the SARS-CoV-2 negative NP samples from the UGA Diagnostic Laboratories, 81% (n = 68/84) were N gene negatives as determined by our RT-qPCR assay. Fluorescent RT-LAMP assay results were 87% (n = 59/68) concordant confirming N gene negative samples, and 16 NP negative samples were positive in our RT-qPCR assay. These results likely represent RT-qPCR false-positives that were SARS-COV-2 negative as determined by the UGA Diagnostic Laboratories with differences likely owing to the broad-specificity of the RT-qPCR assay. The N gene negative NP specimens by RT-LAMP were 13% (n = 9/68) with positive results discordant with RT-qPCR results.

We show that the fluorescent RT-LAMP assay sensitively and specifically detects wild-type SARS-CoV-2 and the B.1.1.7 (Alpha) variant in virus-infected Vero cell culture supernatants after the minimal pre-processing steps. The fluorescent RT-LAMP assay primers were designed based on the nucleotide sequence of the N gene reported for the SARS-CoV-2 (WA1/2020) strain. A majority of the sequence divergence associated with variant strains is located in the S gene, while the N gene nucleotide sequence is highly conserved. The N gene RT-LAMP primer sequences are 100% conserved among the wild type and variant SARS-CoV-2 strains (S2 Fig). This conservation allows our RT-LAMP assay to detect various SARS-CoV-2 strains. Also, the specificity of our fluorescent RT-LAMP assay is shown by the failure of three closely related human coronavirus strains, i.e. OC43, 229E, and NL63, to give any positive signal over the baseline in this 30 minute assay (S2 Fig).

Overall, the fluorescent RT-LAMP assay was shown to be positive for >98% of NP samples (n = 41/42) that were N gene-positive samples by RT-qPCR, and negative for 87% (n = 59/68) of NP samples that were N gene negative samples by RT-qPCR. Discordant results in RT-qPCR and fluorescent RT-LAMP could represent false positives from spurious amplification from primer-dimer sometimes associated with RT-LAMP assays, sample contamination, variability among different RT-qPCR tests detecting different or multiple SARS-CoV-2 amplicons, or increased sensitivity of RT-LAMP compared to RT-qPCR. RT-LAMP employing six primers has high-specificity for selected amplification targets when compared to standard PCR utilizing only two primers, or RT-qPCR employing an additionally labeled probe oligo. Careful design of the six LAMP primers, particularly the inner primers FIP and BIP to avoid self-complementarity, and optimized formulation can help to mitigate these concerns. Our data shows that the fluorescent RT-LAMP assay is comparable to RT-qPCR and useful for detecting SARS-CoV-2 infected NP samples having a moderate viral load. Importantly, the viral load does not indicate either the course of infection or COVID-19 disease severity because the viral load is not representative of the overall viral burden of an infected individual, and the level of SARS-CoV-2 is not the only attribute affecting disease in an individual [37–39].

## Supporting information

**S1 Fig. SARS-CoV-2 WA-1 USA-WA1/2020 detection. A.** RT-qPCR using a CDC EUA N1 probe assay targeting purified genomic RNA from SARS-CoV-2 USA-WA1/2020. Log dilution series of purified genomic viral RNA underwent reverse transcription and cDNA was used in a CDC EUA N1 Probe Assay. Fluorescence data were collected and processed using a 40 cycle cut-off on an AriaMx Real-Time PCR System. **B.** Fluorescent RT-LAMP assay targeting purified SARS-CoV-2 USA-WA1/2020 RNA. Log dilution series of purified genomic RNA ranging from $10^8$ to 10 copies of the N gene was added to fluorescent RT-LAMP reactions containing six LAMP primers (FIP/BIP, F3/B3, and LF/LB) and an SYBR-like fluorescent dye. Reactions were performed on an AriaMx Real-Time PCR System and fluorescence values were plotted against the elapsed reaction time (min) to generate the amplification curve.
(DOCX)

**S2 Fig. SARS-CoV-2 and human coronavirus N gene sequences.** Nucleocapsid (N) gene multiple sequence alignment (T-Coffee) of SARS-CoV-2 strains: SARS2_WA1 (USA-WA1/2020 GenBank: MN985325.1), SARS2_Wuhan1 (Wuhan-Hu-1 GenBank: MN908947), SARS2_VAR_B117 (B.1.1.7 Alpha SARS-CoV-2/human/USA/SEARCH-5574-SAN/2020 GenBank: MW422255.1); and closely-related human coronavirus strains: HuCoV_OC43(GenBank: AY585228.1), HuCoV_NL63 (GenBank: AY567487.2), and HuCoV_229E (GenBank: AF304460.1) used in the study. Position of the six fluorescent RT-LAMP assay primers and their conservation (100%) among the wild-type and variant SARS-CoV-2 strains accounts for assay specificity for the variant strain, SARS2 B.1.1.7. (alpha). Divergence of the heterologous human coronavirus strain sequences results in no signal in the fluorescent RT-LAMP assay for these targets.
(DOCX)

**S3 Fig. SARS-CoV-2 N gene positive control plasmid RT-qPCR.** RT-qPCR using the CDC EUA N1 probe assay targeting SARS-COV-2 N gene positive control plasmid (IDT). Log dilution series of plasmid DNA ranging from $10^5$ to 10 copies of the SARS-CoV-2 N gene used in the CDC EUA N1 Probe Assay. Fluorescence data were collected and processed using a 40 cycle cut-off on an AriaMx Real-Time PCR System and Ct values were plotted against the $Log_{10}$ of the starting N gene copy numbers to generate the standard curve.
(DOCX)

**S1 Table. SARS-CoV-2 fluorescent RT-LAMP primers.** N gene cloning primers, fluorescent RT-LAMP assay primers created using free Primer Explorer V software, and SARS-CoV-2 CDC EUA N1 RT-qPCR assay primers and probe. Numbering according to SARS-CoV-2 Wuhan-Hu-1 (GenBank: MN908947.3).
(DOCX)

**S1 Raw images.**
(PDF)

## Author Contributions

**Conceptualization:** Les Jones, Ralph A. Tripp.

**Data curation:** Les Jones, Ralph A. Tripp.

**Formal analysis:** Ralph A. Tripp.

**Investigation:** Les Jones, Abhijeet Bakre.

**Methodology:** Les Jones, Abhijeet Bakre, Ralph A. Tripp.

**Project administration:** Ralph A. Tripp.

**Resources:** Hemant Naikare, Ravindra Kolhe, Susan Sanchez, Yung-Yi C. Mosley, Ralph A. Tripp.

**Supervision:** Ralph A. Tripp.

**Validation:** Les Jones, Abhijeet Bakre, Ralph A. Tripp.

**Writing – original draft:** Les Jones, Abhijeet Bakre, Yung-Yi C. Mosley, Ralph A. Tripp.

**Writing – review & editing:** Ralph A. Tripp.

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
