## [Decision Letter · Decision Letter 0]

31 May 2021

PONE-D-21-13279

Isothermal Amplification and Fluorescent Detection of SARS-CoV-2 and SARS-CoV-2 Variant Virus in Nasal Swabs

PLOS ONE

Dear Dr. Tripp,

Thank you for submitting your manuscript to PLOS ONE. After careful consideration, we feel that it has merit but does not fully meet PLOS ONE’s publication criteria as it currently stands. Therefore, we invite you to submit a revised version of the manuscript that addresses the points raised during the review process.

Please pay attention to the methods. We have to notice this point in PLOS ONE very much.

We look forward to receiving your revised manuscript.

Kind regards,

Etsuro Ito

Academic Editor

PLOS ONE

Journal Requirements:

2. Thank you for including your ethics statement:  "Samples from consenting adult volunteers undergoing COVID-19 testing and collected anonymously by the Georgia Taskforce for COVID-19, the Georgia National Guard, or from the University of Georgia Diagnostic Laboratory. ".   

Please amend your current ethics statement to confirm that your named institutional review board or ethics committee specifically approved this study.

6. We note you have included a table to which you do not refer in the text of your manuscript. Please ensure that you refer to Tables 2-4 in your text; if accepted, production will need this reference to link the reader to the Table.

Reviewers' comments:

Reviewer's Responses to Questions

**Comments to the Author**

1. Is the manuscript technically sound, and do the data support the conclusions?

Reviewer #1: Yes

Reviewer #2: Partly

Reviewer #3: Yes

Reviewer #4: Yes

2. Has the statistical analysis been performed appropriately and rigorously? 

Reviewer #1: N/A

Reviewer #2: No

Reviewer #3: Yes

Reviewer #4: I Don't Know

3. Have the authors made all data underlying the findings in their manuscript fully available?

Reviewer #1: Yes

Reviewer #2: No

Reviewer #3: Yes

Reviewer #4: Yes

4. Is the manuscript presented in an intelligible fashion and written in standard English?

Reviewer #1: Yes

Reviewer #2: No

Reviewer #3: Yes

Reviewer #4: Yes

5. Review Comments to the Author

Reviewer #1: The authors have documented the development of a Fluorescent LAMP detection method that can be completed in 30 minutes from minimally pre-processed nasal swabs. It shows to be faster and sensitive enough, when compared to RT-PCR, the globally accepted standard technique for detecting SARS-CoV-2.

Repeated information in Introduction second paragraph: “reduced cost, speed…sensitivity”. It can be consolidated.

Materials and Methods, RT-LAMP reactions, second paragraph:

Please specify which kind of sample in: “For patient sample fluorescent…”

The micro units must be spelled correctly as symbol, not “u”.

Same paragraph as above, word specificity is misspelled.

Fourth paragraph: …”QPCR machine”…, use thermal cycler or thermocycler.

Tables 2 through 4 are not cited in the main text. This needs to be corrected.

One important issue addressed in this paper is that the proposed method has the capacity to detect different SARS-CoV-2 variants. It would help the paper to (briefly) discuss why that is technically possible.

Reviewer #2: This work is part of a series of studies that aim to develop precise and fast methods to detect the SARS-CoV2 virus in patient samples. The objective of the study is to develop a diagnostic test that can be completed in 30 min without isolating RNA from the samples using Fluorescence as a read-out.

The novel finding in this study is that their method has the potential to work with more than one virus variant. RT-LAMP methods that have no prior RNA isolation have been published as well as RT-LAMP using fluorescence as read-out. For further references review Thompson et al., 2020 https://doi.org/10.1016/j.snr.2020.100017. Compared to other published methods the results shown in this article show low sensitivity and specificity.

The data presented should not be separated according to the RT-qPCR Ct’s. When testing patients we cannot know which are going to have a higher or lower viral load, therefore sensitivity and specificity of the (whole) method should be given.

The article needs to improve the language quality and has almost no mention of other current RT-LAMP SARS-CoV2 detection methods. Comparing the proposed RT-LAMP fluorescence diagnostic test against RT-qPCR makes sense since it is the standard diagnostic test, but adding a full comparison seems appropriate as well as some emphasis on why do we need more options. The discussion should address better why is this method better or different.

Major revisions.

1. The abstract lacks the main findings of the paper and a small conclusion. A clear explanation on why do we need a better or different method should be added.

2. To date, there have been over 163 million cases and over 3 million deaths worldwide. What is “CONSIDERABLE morbidity and mortality” and how does it “affect 12 million people” This data has no reference so I couldn’t find what the affected 12 million people were referring to. All data should have a reference. An update in the number of cases can be found here: https://coronavirus.jhu.edu/map.html

3. It is highly recommended to have an English editor. The document has numerous grammar mistakes and lacks the general flow of the ideas from general to particular.

4. The introduction needs to be focused on the article’s topic. The first paragraph starts with the origin of the coronavirus, then statistics, pathogenic mechanisms, viral transmission, symptoms and vaccines. A lot of subjects are mentioned in a very superficial way. The description of the interaction between the S1 Unit of the S protein of the virus with the ACE2 of the cell is very confusing and adds no value to the paper. It would be better to have a well explained current testing methods focused introduction that really introduces the paper and has one clear goal: why is developing new testing methods necessary?

Many articles on RT-LAMP SARS-CoV2 detection methods with no prior RNA isolation have been published and there is mention of any of these in the introduction. RT-LAMP methods that use fluorescence as readout have also been published before. Since none of this is novel, they should be mentioned.

If there are any issues in the current LAMP detection methods, how common are these? How does it affect sensitivity? Why would we want to change them? All references used here are with other virus, there are articles that test SARS-CoV2 with RT-LAMP.

5. Reorganizing and adding more subtitles in the method section could make it easier to understand. Start with the SARS-CoV-2 RNA, IVT RNA and how were patient samples obtained headings. Then explain the treatments.

Separate the protocols for Vero cells infections, RT-qPCR and LAMP in Vero cells and then Patient samples treatment: RT-qPCR with RNA isolation and LAMP. Try not to add results in the method section.

6. Add a statement on if the samples where obtained with IRB permission and maintained in a deidentified manner, etc

7. The LOD does not determine the sensitivity of the test.

8. Why are some assays and controls not shown? They can be added to the Supplementary files.

9. Results are very hard to understand. Clustering sensitivity and specificity of each sample type might help: Address first the differences between the RT-qPCR from the official labs and the RT-qPCR done in your lab. It would also be helpful to add different names or an extra letter to differentiate this. Then address the sensitivity and specificity of the RT-LAMP.

The text does not match the Tables: For table 1, where are the negative samples? Or how did we get the 81% indicated in the text? Tables 2 and 3 are not referenced in the text.

10. Why are the tables divided depending on the Ct values? And why is this not in the text? Separating the data according to Ct values makes the RT-LAMP look good, I understand it could be interesting to mention that in the discussion, but the sensitivity and specificity of the method should include all samples.

11. According to the method section, there are 50 positive samples and 84 negative ones. Table 1 and 2 add up for the 50 positive samples. Why are there only 16 negative ones in table 3 and why do they have a Ct if they are negative?

12. Discussion should be focused on comparing this method with the other available methods.

Minor comments.

1. Real time reverse transcription PCR is abbreviated RT-qPCR not RT-PCR or qRT-PCR. Make this consistent in all the text.

2. “SARS-CoV-2 is an RNA virus with a genome that encodes four (MAIN?) proteins...” I would add the main because in the same sentence it says it encodes other proteins.

3. Do not number the reagents in a PCR/LAMP reaction; write the methods in a scientific way. Also, number 4 is repeated in the RT-LAMP setup reaction.

4. Change the u to μ.

5. What liquid sample are we talking about in the second paragraph of the Materials & Methods?

6. In methods: change -“soak” to inactivate Proteinase K- to –incubate-. Check for spelling minor mistakes (specicity).

7. Change Cq values to Ct

8. The sentence: “The viral load in SARS-CoV-2 patients is reported to range from 641 copies/mL to 1011 copies/mL with a median of 104 – 105” doesn’t make sense.

9. Forgot some () when referring to Figures in the text, and check how to cite the Figures in the text according to the Plos publishing guidelines.

10. Add confidence intervals to statistics.

11. Change the Ct value to ranges. Ct values < 40 include form 0 to 40. A range from 31-40 would be appropriate.

13. On the figures: Check that numbers and letters match. Each figure should have a title in the figure legend.

• Table: RT-qPCR

• On table 3. There is a Ct value of 16 in sample 12. Is this correct?

Reviewer #3: This article describes an RNA amplification detection method using loop-mediated isothermal amplification (LAMP) to achieve specific, rapid (30 min) and sensitive. There are few major comments that the authors should provide more evidence and justification.

1. There are two qRT-PCR methods used in this study, the authors need to clarify the differences and name it differently to avoid confusion.

2. Analytical sensitivity of 100 copies is not high as compared to RT-LAMP of previous studies, this has to be justified in ‘Discussion’.

3. SARS-CoV-2 positive samples with qRTPCR Ct values >30 were only 65% (n = 13/20) detected by fluorescent RT-LAMP assay. Better explanation must be provided to justify the low sensitivity and possible ways of improvement.

3. Similarly for false positive results, proper reasons should be provided and ways to avoid contamination.

5. The authors sought to improve upon SARS-CoV-2 diagnostic testing by reducing the time and costs to acquire the results. However, in this assay, a AriaMX Real-Time PCR System machine us still needed which is costly.

Reviewer #4: Comments to the authors.

In the presented manuscript the authors developed a novel assay employing a LAMP method to detect SARS-CoV-2 and SARS-CoV-2 variant virus. In this COVID-19 pandemic, LAMP method can be alternative to qPCR. Therefore, this study is very valuable. However, this article possesses some technical shortcomings and requires considerable revision as listed below.

Major comments are as follows:

1. Authors must obtain IRB approval and show IRB approval number in the manuscript.

2. Abstract should include actual data in this study. (e.g. Sensitivity and specificity of LAMP method in clinical samples.)

3. Authors should confirm whether LAMP products match with the expected sequences by direct sequencing.

4. Authors should discuss the accuracy of the LAMP method and qPCR using sensitivity and specificity referring to the CDC 2019-nCov qPCR assay. A 2×2 table may help readers to understand the accuracy of each method. (e.g. Table 1 in J Med Microbiol. 2015 Nov;64(11):1335-1340.)

5. Authors should show the easily understandable interpretation of Table 1-4.

If they discuss sensitivity, specificity, and false positives, a 2×2 table is easier to understand. If they discuss the viral load, Table 1-4 is indispensable.

6. Authors should avoid including the discussions in the result.

・The viral load in SARS-CoV-2 patients is reported to range from 641 copies/mL to 1011 copies/mL with a median of 104 - 105 [31].

・Form ‘Importantly’ to ‘when the Ct value was <30 and >40 for negative fluorescent RT-LAMP result’.

Minor comments are as follows:

7. The title of Figure 2 is the same as that of figure 3. Authors should set appropriate titles.

8. In the legend of figure 3. A, B between ‘and’ and ‘Fluorescent RT-LAMP assay’ is a typo?

9. Authors should confirm the titles of Figure 4.

The title of A is SARS CoV-2 WA-1 Cell culture lysate?

The title of B is SARS CoV-2 Variant B. 1. 1. 7 variant cell culture lysate?

6. PLOS authors have the option to publish the peer review history of their article (what does this mean?). If published, this will include your full peer review and any attached files.

Reviewer #1: **Yes: **Jorge A. Huete-Pérez

Reviewer #2: No

Reviewer #3: No

Reviewer #4: No

---

## [Author Response · Author response to Decision Letter 0]

30 Jun 2021

PONE-D-21-13279: Isothermal Amplification and Fluorescent Detection of SARS-CoV-2 and SARS-CoV-2 Variant Virus in Nasal Swabs

Reviewer #1: The authors have documented the development of a Fluorescent LAMP detection method that can be completed in 30 minutes from minimally pre-processed nasal swabs. It shows to be faster and sensitive enough, when compared to RT-PCR, the globally accepted standard technique for detecting SARS-CoV-2.

Repeated information in Introduction second paragraph: “reduced cost, speed…sensitivity”. It can be consolidated.

• We have consolidated the Introduction as suggested. 

Materials and Methods, RT-LAMP reactions, second paragraph:

Please specify which kind of sample in: “For patient sample fluorescent…”

• The sentence was amended to “For fluorescent RT-LAMP reactions of NP swabs…” 

The micro units must be spelled correctly as symbol, not “u”.

• This has been corrected. 

Same paragraph as above, word specificity is misspelled.

• This has been corrected. 

Fourth paragraph: …”QPCR machine”…, use thermal cycler or thermocycler.

• The words, “real-time thermocycler” has been substituted. 

Tables 2 through 4 are not cited in the main text. This needs to be corrected.

• This has been corrected. Results Section.

One important issue addressed in this paper is that the proposed method has the capacity to detect different SARS-CoV-2 variants. It would help the paper to (briefly) discuss why that is technically possible.

• We agree and have added pertinent information to the Discussion Section. 

• We now include Supplemental Figure 2 showing SARS-CoV-2, SARS-CoV-2 variant and human coronavirus N gene sequences regarding detection.

Reviewer #2: This work is part of a series of studies that aim to develop precise and fast methods to detect the SARS-CoV2 virus in patient samples. The objective of the study is to develop a diagnostic test that can be completed in 30 min without isolating RNA from the samples using Fluorescence as a read-out.

The novel finding in this study is that their method has the potential to work with more than one virus variant. RT-LAMP methods that have no prior RNA isolation have been published as well as RT-LAMP using fluorescence as read-out. For further references review Thompson et al., 2020 https://doi.org/10.1016/j.snr.2020.100017. Compared to other published methods the results shown in this article show low sensitivity and specificity.

• We appreciate the reviewer’s comments but disagree that our manuscript shows low sensitivity and specificity. We show that the RT-LAMP reactions are specific and sensitive to <100 copies by fluorescent detection. We show that the fluorescent RT-LAMP limit of detection for SARS-CoV-2 N gene RNA is ~100 copies which compares favorably to the limit of detection of the CDC EUA N1 RT-PCR assay which is ~10 copies for the same templates. In our studies, 100 copies of N gene corresponds to a limit of detection of 0.1 fg of RNA in our LAMP assay which is commensurate with other published SARS-CoV-2 LAMP studies.

The data presented should not be separated according to the RT-qPCR Ct’s. When testing patients we cannot know which are going to have a higher or lower viral load, therefore sensitivity and specificity of the (whole) method should be given.

The article needs to improve the language quality and has almost no mention of other current RT-LAMP SARS-CoV2 detection methods. Comparing the proposed RT-LAMP fluorescence diagnostic test against RT-qPCR makes sense since it is the standard diagnostic test, but adding a full comparison seems appropriate as well as some emphasis on why do we need more options. The discussion should address better why is this method better or different.

• We have revised the Tables to show the overall results of the RT-LAMP analysis of NP samples and explain potential differences between RT-qPCR sensitivity and how this may impact RT-LAMP when comparing to RT-qPCR on the same NP samples. 

• We focus the discussion on RT-LAMP assay detection of SARS-CoV-2 without RNA extraction as most of the literature on RT-LAMP uses purified RNA or synthetic DNA templates as RT-LAMP targets which is not the focus of this manuscript.

• We clarify in the revised manuscript the rationale for RT-LAMP fluorescent detection.

Major revisions.

1. The abstract lacks the main findings of the paper and a small conclusion. A clear explanation on why do we need a better or different method should be added.

• We have amended the abstract to succinctly provide conclusions, and note that “…RT-LAMP has several advantages over RT-qPCR including isothermal amplification, speed (<30 min), reduced costs, and similar sensitivity and specificity”. 

2. To date, there have been over 163 million cases and over 3 million deaths worldwide. What is “CONSIDERABLE morbidity and mortality” and how does it “affect 12 million people” This data has no reference so I couldn’t find what the affected 12 million people were referring to. All data should have a reference. 

• The statement has been edited to note that “According to the WHO Coronavirus (COVID-19) Detailed Surveillance Data Dashboard, currently there are more than 170 million confirmed cases, and more than 3.8 million deaths from COVID-19 worldwide”.

3. It is highly recommended to have an English editor. The document has numerous grammar mistakes and lacks the general flow of the ideas from general to particular.

• We apologize for any errors - they have been rectified.

4. The introduction needs to be focused on the article’s topic. The first paragraph starts with the origin of the coronavirus, then statistics, pathogenic mechanisms, viral transmission, symptoms and vaccines. A lot of subjects are mentioned in a very superficial way. The description of the interaction between the S1 Unit of the S protein of the virus with the ACE2 of the cell is very confusing and adds no value to the paper. It would be better to have a well explained current testing methods focused introduction that really introduces the paper and has one clear goal: why is developing new testing methods necessary?

• We believe the introduction properly addresses the topic of the manuscript, and that a level of discussion regarding the biology of SARS-CoV-2 is needed. We do discuss some of the current RT-LAMP methods in the Introduction as suggested. In the Abstract, we note that ‘Although various nucleic acid-based assays have been developed to aid the detection of SARS-CoV-2 from COVID-19 patient samples, the objective of this study was to develop a diagnostic test that can be completed in 30 minutes without having to isolate RNA from the samples.’ Then in the first paragraph of the Introduction, we present why we are developing a diagnostic assay for SAR-CoV-2 and what is being done to address the problem. 

Many articles on RT-LAMP SARS-CoV2 detection methods with no prior RNA isolation have been published and there is mention of any of these in the introduction. RT-LAMP methods that use fluorescence as readout have also been published before. Since none of this is novel, they should be mentioned.

• We are unaware of many articles published on fluorescent RT-LAMP detection of SARS-CoV-2 directly from clinical samples. We believe we have suitably referenced related RT-LAMP manuscripts most which use purified RNA or in vitro transcribed RNA, or synthetic DNA-based SARS-CoV-2 templates for detection.

If there are any issues in the current LAMP detection methods, how common are these? How does it affect sensitivity? Why would we want to change them? All references used here are with other virus, there are articles that test SARS-CoV2 with RT-LAMP.

• We appreciate the reviewer’s questions and note in the Discussion section that discordant results in RT-qPCR and fluorescent RT-LAMP could represent false-positives from spurious amplification from primer-dimers, sample contamination, variability among different RT-qPCR tests detecting different or multiple SARS-CoV-2 amplicons, or increased sensitivity of RT-LAMP compared to RT-qPCR. RT-LAMP employing six primers should have relative high-specificity for selected amplification targets when compared to standard PCR utilizing only two primers or RT-qPCR employing an additional labeled probe oligo. Careful design of the six RT-LAMP primers, particularly the inner primers FIP and BIP, to avoid self-complementarity, and optimized formulation can help to mitigate these concerns. 

5. Reorganizing and adding more subtitles in the method section could make it easier to understand. Start with the SARS-CoV-2 RNA, IVT RNA and how were patient samples obtained headings. Then explain the treatments.

Separate the protocols for Vero cells infections, RT-qPCR and LAMP in Vero cells and then Patient samples treatment: RT-qPCR with RNA isolation and LAMP. Try not to add results in the method section.

• We agree and have re-organized the Materials & Methods section to address the reviewer’s suggestions and have added additional headings.

6. Add a statement on if the samples where obtained with IRB permission and maintained in a deidentified manner, etc

• The deidentified NP samples were collected from consenting adult volunteers under 45 CFR 46.102(l)(2) (the Common Rule) as a public health surveillance activity by the Georgia Taskforce for COVID-19, the Georgia National Guard, and the University of Georgia Diagnostic Laboratory. 

7. The LOD does not determine the sensitivity of the test.

• We determined the analytical sensitivity by an end-point dilution to the point that the RT-LAMP assay can no longer is able detect the target. We note that the LOD for our RT LAMP assay using SARS-COV-2 N gene RNA is ~100 copies of N gene corresponding to 1fg of RNA at 95% confidence interval. Our current RT-LAMP assay is highly specific for SARS-CoV-2 as it did not give a positive result with other closely-related human coronaviruses (NL63, 229E, and OC43). 

8. Why are some assays and controls not shown? They can be added to the Supplementary files.

• The control assay data is now included in the Results section and Figure 3.

9. Results are very hard to understand. Clustering sensitivity and specificity of each sample type might help: Address first the differences between the RT-qPCR from the official labs and the RT-qPCR done in your lab. It would also be helpful to add different names or an extra letter to differentiate this. Then address the sensitivity and specificity of the RT-LAMP.

The text does not match the Tables: For table 1, where are the negative samples? Or how did we get the 81% indicated in the text? Tables 2 and 3 are not referenced in the text.

• We have redone the Tables in the revised manuscript to provide clarity. The University of Georgia Diagnostic Laboratories confirmed the SARS-CoV-2 status of the NP samples used in our analysis employing the “ThermoFisher Applied Biosystems TaqPath COVID-19 EUA Combo kit” to detect SARS-CoV-2 RNA in a multiplex qRT-PCR format. This indicated in the revised manuscript. The assay is specific for three SARS-CoV-2 genome regions: S gene, N gene, and orf-1ab. We analyzed the same NP amples using the CDC EUA N1 qRT-PCR assay for SARS-CoV-2 RNA. The CDC EUA N1 assay is specific for the SARS-CoV-2 nucleocapsid gene only. We reported Ct values generated from our analysis of patient NP swab samples by the CDC EUA N1 qRT-PCR assay. 

• We determined the presence of SARS-CoV-2 directly in clinical NP swab samples collected from patients using fluorescent RT-LAMP. Nasal swabs from suspected COVID-19 patients were collected by personnel from the Georgia COVID-19 Task Force and The University of Georgia Diagnostic Laboratories and confirmed positive or negative for SARS-CoV-2 using a multiplex RT-qPCR assay as described above. When comparing fluorescent RT-LAMP to RT-qPCR, we chose the singleplex N gene-specific SARS-CoV-2 CDC EUA N1 RT-qPCR assay since it shares close target specificity with the fluorescent RT-LAMP assay (Supplemental Table 1). Using the same NP samples, we performed the CDC EUA N1 RT-qPCR assay on both SARS-CoV-2 positive (n=50) and SARS-CoV-2 negative (n=84) NP specimens in triplicate. Positive samples having CDC EUA N1 RT-qPCR probe assay Ct values <40 for each of three repeats were considered N gene positives for the comparative analysis. We observed that 84% (n=42/50) of the previously confirmed positive SARS-CoV-2 NP specimens produced a matching N gene positive result (Ct<40) using the CDC EUA N1 RT-qPCR assay for each of three repeat measurements (Table 1). The mean Ct for the SARS-CoV-2 N gene positive NP samples (n=42) ranged from 16 to 34 cycles, with an overall mean Ct of 25.7, 95% CI [24.02 - 27.39] using the SARS-CoV-2 CDC EUA N1 RT-qPCR assay. Negative samples with a (No Ct) result in the CDC EUA N1 RT-qPCR assay for each of three repeat measurements were considered N gene negatives. Of the previously confirmed SARS-CoV-2 negative NP samples, we observed that (n=68/84) 81% produced a matching N gene negative result (No Ct) in three repeat measurements with the SARS-CoV-2 CDC EUA N1 RT-qPCR assay (Table 2). 

• The fluorescent RT-LAMP assay was performed in triplicate on the same clinical NP specimens that were previously determined to be positive for SARS-CoV-2 (n=50) by the University of Georgia Diagnostic Laboratories. The mean fluorescent RT-LAMP assay values (minutes) for the SARS-CoV-2 N gene positive NP samples giving a positive RT-LAMP signal (n=41) ranged from 18 to 28 minutes, with an overall mean minutes of 23.4, 95% CI [22.69 - 24.09]. Results for the fluorescent RT-LAMP assay on the N gene positive NP samples (n=42), showed 98% (n=41/42) concordance with RT-qPCR for SARS-CoV-2 positivity (Table 1). Non N gene positive NP samples, signified by (No Ct) result in the CDC EUA N1 RT-qPCR assay (8/50) had discordant fluorescent RT-LAMP results signified by positive signal in the RT-LAMP assay (n=7/8). 

• Fluorescent RT-LAMP was performed on the same NP samples that were previously shown to be SARS-CoV-2 negative (n=84) by the University of Georgia Diagnostic Laboratories. Results for the fluorescent RT-LAMP assay on the N gene negative NP samples (n=68) showed 87% (n = 59/68) concordance with RT-qPCR for SARS-CoV-2 negativity (Table 2), while 13% (n=9/68) of N gene negative samples gave discordant fluorescent RT-LAMP assay results signified by a positive result in the 30 minute assay time.

• When compared to RT-qPCR, N gene positive samples assayed by fluorescent RT-LAMP displayed 98% (n = 41/42) concordance, while fluorescent RT-LAMP values for SARS-CoV-2 N gene negative samples were 87% (n = 59/68) concordant with RT-qPCR. Importantly, the fluorescent RT-LAMP assay results were obtained without purification of RNA from the NP specimens in contrast to the RT-qPCR assay protocol. RESULTS Section

10. Why are the tables divided depending on the Ct values? And why is this not in the text? Separating the data according to Ct values makes the RT-LAMP look good, I understand it could be interesting to mention that in the discussion, but the sensitivity and specificity of the method should include all samples.

• The tables have been reorganized to better reflect the comparisons made. Results Section and Tables 1 and 2. 

11. According to the method section, there are 50 positive samples and 84 negative ones. Table 1 and 2 add up for the 50 positive samples. Why are there only 16 negative ones in table 3 and why do they have a Ct if they are negative? 

Revised in Discussion Section:

• Of the original group of SARS-CoV-2 positive (n=50) NP samples, 84% (n=42/50) were true N gene positives as determined by RT-qPCR assay. Fluorescent RT-LAMP assay results were 98% (n=41/42) concordant on these same N gene positive specimens. 8 SARS-CoV-2 positive NP samples, as determined by the University of Georgia Diagnostic Laboratories, failed to amplify the N gene target in our RT-qPCR assay. However, the internal control amplicon (human RNAseP) amplified competently in all NP specimens we tested indicating that extracted RNA was intact. N=7/8 of the failed RT-qPCR NP specimens resulted in a positive fluorescent RT-LAMP assay discordant with our RT-qPCR result for these NP samples, but in agreement with the original SARS-CoV-2 positive status confirmed by the University of Georgia Diagnostic Laboratories. This may indicate sample preparation differences pursuant to running the assays (RNA purification for RT-qPCR or minimal pre-processing for RT-LAMP) contributes to different results. We show that the fluorescent RT-LAMP assay sensitivity is more than sufficient to reliably detect SARS-CoV-2 in patient NP samples as the viral load in SARS-CoV-2 patients is reported to range from 641 copies/mL to 1011 copies/mL with a median of 104 - 105 [31]. Of the original group of SARS-CoV-2 negative (n=84) NP specimens, 81% (n=68/84) were N gene negatives as determined by our RT-qPCR assay. Fluorescent RT-LAMP assay results were 87% (n=59/68) concordant on these same true N gene negative specimens. 16 SARS-CoV-2 negative NP samples, as determined by the University of Georgia Diagnostic Laboratories, gave a positive result in our RT-qPCR assay. These results could represent RT-qPCR false-positives in our assay, but were clearly determined SARS-COV-2 negative by the University of Georgia Diagnostic Laboratories with the difference likely owing to their broad-specificity in the RT-qPCR assay. Fluorescent RT-LAMP, with respect to the N gene negative NP samples gave 13% (n=9/68) with positive results discordant with RT-qPCR. 

12. Discussion should be focused on comparing this method with the other available methods.

• We compared the ‘gold-standard’ qRT-PCR to the fluorescent RT-LAMP assay and discuss performance in detecting SARS-CoV-2 from the same clinical samples as that was the goal of the this study. Discussion Section.

Minor comments.

1. Real time reverse transcription PCR is abbreviated RT-qPCR not RT-PCR or qRT-PCR. Make this consistent in all the text.

• This is corrected throughout the text.

2. “SARS-CoV-2 is an RNA virus with a genome that encodes four (MAIN?) proteins...” I would add the main because in the same sentence it says it encodes other proteins.

• This has been changed to four “structural” proteins. Introduction section

3. Do not number the reagents in a PCR/LAMP reaction; write the methods in a scientific way. Also, number 4 is repeated in the RT-LAMP setup reaction.

• This is corrected in Materials and Methods section. Numbering has been removed.

4. Change the u to μ.

• This is corrected in the revised manuscript. 

5. What liquid sample are we talking about in the second paragraph of the Materials & Methods?

• This is the nasopharyngeal (NP) swab samples which has been clarified throughout the revised manuscript.

6. In methods: change -“soak” to inactivate Proteinase K- to –incubate-. Check for spelling minor mistakes (specicity).

• This has been corrected.

7. Change Cq values to Ct

• This is corrected.

8. The sentence: “The viral load in SARS-CoV-2 patients is reported to range from 641 copies/mL to 1011 copies/mL with a median of 104 – 105” doesn’t make sense.

• The sentence has been amended in the Discussion section to provide context within the RT-LAMP reaction results. We show that the fluorescent RT-LAMP assay sensitivity is more than sufficient to reliably detect SARS-CoV-2 in NP samples with respect to virus load, as SARS-CoV-2 patients have been reported to have a range from 641 copies/mL to 1011 copies/mL with a median of 104 - 105 [31]. 

9. Forgot some () when referring to Figures in the text, and check how to cite the Figures in the text according to the Plos publishing guidelines.

• This is corrected.

10. Add confidence intervals to statistics.

• 95% CI are now added to Tables 1 and 2 and in Results section referring to same. 

11. Change the Ct value to ranges. Ct values < 40 include form 0 to 40. A range from 31-40 would be appropriate.

• We agree and include the Ct ranges in Tables 1 and 2 and in Results section.

13. On the figures: Check that numbers and letters match. Each figure should have a title in the figure legend.

• Thank you for bringing that to our attention. It is corrected.

On table 3. There is a Ct value of 16 in sample 12. Is this correct?

• Yes, the Ct for this sample = 16. 

Reviewer #3: This article describes an RNA amplification detection method using loop-mediated isothermal amplification (LAMP) to achieve specific, rapid (30 min) and sensitive. There are few major comments that the authors should provide more evidence and justification.

1. There are two qRT-PCR methods used in this study, the authors need to clarify the differences and name it differently to avoid confusion.

• We concur and have amended the manuscript to provide clarity. The University of Georgia Diagnostic Laboratory confirmed the SARS-CoV-2 status of the patient nasopharyngeal swab (NP) samples used in our analysis employing the “ThermoFisher Applied Biosystems TaqPath COVID-19 EUA Combo Kit” to detect SARS-CoV-2 RNA in a multiplex qRT-PCR format. The assay is specific for three SARS-CoV-2 genome regions: S gene, N gene, and orf-1ab. We analyzed the same NP specimens using the CDC EUA N1 qRT-PCR assay for SARS-CoV-2 RNA. The CDC EUA N1 assay is specific for the SARS-CoV-2 Nucleocapsid gene only. Materials and Methods section.

2. Analytical sensitivity of 100 copies is not high as compared to RT-LAMP of previous studies, this has to be justified in ‘Discussion’.

• We show that the fluorescent LAMP limit of detection for SARS-CoV-2 N gene RNA is ~100 copies which compares favorably to the limit of detection of the CDC EUA N1 qRT-PCR assay which is ~10 copies for the same templates. In our studies, 100 copies of N gene corresponds to a limit of detection of 1 fg of RNA in our RT-LAMP assay which is commensurate with other published SARS-CoV-2 RT-LAMP studies. Results and Discussion sections

3. SARS-CoV-2 positive samples with qRTPCR Ct values >30 were only 65% (n = 13/20) detected by fluorescent RT-LAMP assay. Better explanation must be provided to justify the low sensitivity and possible ways of improvement.

• We have revised the Tables and clarified the results. The Ct values generated were from our analysis of patient NP swab samples by the CDC EUA N1 qRT-PCR assay. We analyzed n=50 NP specimens confirmed SARS-CoV-2 positive by the University of Georgia Diagnostic Laboratories who employed a multiplex qRT-PCR assay with specificity for 3 SARS-CoV-2 genomic regions. Since the CDC EUA N1 assay that we used is a singleplex assay with specificity only for the SARS-CoV-2 N gene, we would expect to see reduced overall SARS-CoV-2 sensitivity in patient NP samples when compared to the multiplex assay. Our fluorescent RT-LAMP assay also has singleplex specificity for the SARS-CoV-2 N gene, and we would expect similar results. Although these n=50 NP specimens were confirmed SARS-CoV-2 positive using a multiplex assay, our analysis indicates that the N gene templates are in relative low abundance likely via sample degradation as evidenced by high N gene-specific Ct values (Ct>30), and this is reflected in the results for our N gene-specific RT-LAMP assay on the same samples. Results section 

3. Similarly for false positive results, proper reasons should be provided and ways to avoid contamination.

• We concur and have added additional discussion. Briefly, it is understood that RT-LAMP diagnostics can produce false-positive results, and that detailed understanding of the phenomenon is lacking in the literature. Presumably, RT-LAMP employing six primers should have relative high-specificity for selected amplification targets when compared to standard PCR utilizing only two primers or qRT-PCR employing an additional labeled probe oligo. False-positive results associated with RT-LAMP have been reported to result from primer-dimer elongation or spurious amplification. We believe that careful design of the six LAMP primers, particularly the inner primers FIP and BIP, to avoid self-complementarity, and optimized formulation can help to mitigate these concerns. Discussion section 

5. The authors sought to improve upon SARS-CoV-2 diagnostic testing by reducing the time and costs to acquire the results. However, in this assay, a AriaMX Real-Time PCR System machine us still needed which is costly.

• We agree with the reviewer that the AriaMx machine is costly equipment and itself does not lead to reduced assay costs. However, in this manuscript we show the feasibility of substituting RT-LAMP for qRT-PCR analysis of clinical specimens by showing that RT-LAMP has similar specificity and sensitivity for SARS-CoV-2 detection in a side-by-side comparison. Further, we note that specimens in the RT-LAMP protocol are analyzed with minimal processing having not undergone expensive and time consuming RNA purification required for qRT-PCR. We show that RT-LAMP can meet the requirements for screening and testing for SARS-CoV-2 and potentially be a complementary tool to qRT-PCR. Discussion section

Reviewer #4: Comments to the authors.

In the presented manuscript the authors developed a novel assay employing a LAMP method to detect SARS-CoV-2 and SARS-CoV-2 variant virus. In this COVID-19 pandemic, LAMP method can be alternative to qPCR. Therefore, this study is very valuable. However, this article possesses some technical shortcomings and requires considerable revision as listed below.

Major comments are as follows:

1. Authors must obtain IRB approval and show IRB approval number in the manuscript.

• As noted all deidentified NP samples were collected from consenting adult volunteers under 45 CFR 46.102(l)(2) (the Common Rule) as a public health surveillance activity by the Georgia Taskforce for COVID-19, the Georgia National Guard, and the University of Georgia Diagnostic Laboratory. 

2. Abstract should include actual data in this study. (e.g. Sensitivity and specificity of LAMP method in clinical samples.)

• We are limited by the journal from, expanding the abstract and therefore included selected data in the abstract.

3. Authors should confirm whether LAMP products match with the expected sequences by direct sequencing.

• We agree that direct sequencing of the RT-LAMP products would confirm specificity, however we do not believe it is necessary. All negative control RT-LAMP assay targets including heterologous human coronaviruses (NL63, OC43, and 229E) failed to produce detectable amplification products in the SARS-CoV-2 RT-LAMP assay. Only closely-related SARS-CoV-2 strains, WA1 and variant strain B.1.1.7 (all sharing highly-conserved N gene sequence) generated concentration-dependent signal across a wide range of target concentrations in the LAMP assay. Supplementary Figure 2

4. Authors should discuss the accuracy of the LAMP method and qPCR using sensitivity and specificity referring to the CDC 2019-nCov qPCR assay. A 2×2 table may help readers to understand the accuracy of each method. (e.g. Table 1 in J Med Microbiol. 2015 Nov;64(11):1335-1340.)

• We discuss sensitivity and specificity as measures of the proportion of positives or negatives that are correctly identified by RT-LAMP compared to RT-PCR. We re-organized the data into two Tables for ease of understanding. Patient samples were previously confirmed as SARS-CoV-2 positive or negative by the University of Georgia Diagnostic Laboratories using the ThermoFisher Applied Biosystems TaqPath COVID-19 EUA Combo Kit to detect SARS-CoV-2 RNA in a multiplex RT-qPCR format specific for three SARS-CoV-2 genome regions: S gene, N gene, and orf-1ab. We further analyzed the same NP specimens by RT-qPCR using the singleplex CDC EUA N1 probe assay with a cut-off of 40 cycles, and our N gene-specific fluorescent RT-LAMP assay with a cut-off at 30 minutes. FIP/BIP, F3/B3, and LF/LB primers and an SYBR-like fluorescent dye were used in the RT-LAMP assay. The SARS-CoV-2 positive sample fluorescent RT-LAMP results (n=50) were grouped according to their CDC EUA N1 RT-qPCR assay result: specimens below the assay cut-off (Ct<40) n=42/50 or, specimens above the assay cut-off (No Ct) n=8/50. The number of positive specimens having a fluorescent RT-LAMP assay value less than the 30 minute assay cut-off (n=41/42) have concordant results to RT-qPCR for these specimens. One specimen (n=1/42) had discordant results in fluorescent RT-LAMP assay as it failed to produce a signal above background in the 30 minute assay time. The average fluorescent RT-LAMP assay value (23 minutes) is calculated for the group of positive specimens (n=41) at the 95%CI, 23 [22.6- 24.09]. Specimens having (No Ct) result in RT-qPCR (8/50) had discordant fluorescent RT-LAMP results signified by positive signal in the RT-LAMP assay (n=7/8). Fluorescent RT-LAMP is measured by the time in minutes required to generate fluorescence signal (ΔR) over baseline in the assay. 

• Table 2. Patient samples were previously confirmed as SARS-CoV-2 positive or negative by the University of Georgia Diagnostic Laboratories using the ThermoFisher Applied Biosystems TaqPath COVID-19 EUA Combo Kit to detect SARS-CoV-2 RNA in a multiplex RT-qPCR format specific for three SARS-CoV-2 genome regions: S gene, N gene, and orf-1ab. We further analyzed the same NP specimens by RT-qPCR using the singleplex CDC EUA N1 probe assay with a cut-off of 40 cycles, and our N gene-specific fluorescent RT-LAMP assay with a cut-off at 30 minutes. FIP/BIP, F3/B3, and LF/LB primers and an SYBR-like fluorescent dye were used in the RT-LAMP assay. The SARS-CoV-2 negative sample fluorescent LAMP results (n=84) were grouped according to their CDC EUA N1 RT-qPCR assay result: specimens showing No Ct (n=68/84) or, specimens having Ct below the assay cut-off Ct<40 (n=16/84). N =68 of the negative NP specimens failed to signal in our RT-qPCR assay, as expected. Of this group, n=59/68 gave concordant fluorescent RT-LAMP assay results failing to signal above baseline in the 30 minute assay time, and n=9/68 gave discordant fluorescent RT-LAMP assay results signified by a positive fluorescent LAMP signal in the 30 minute assay time. Some of the negative NP specimens gave RT-qPCR Ct values below the assay cut-off (n=16/84) suggesting a positive result. This is unexpected from negative specimens, but likely owes to the differential specificities between our singleplex CDC EUA N1 RT-qPCR assay and the multiplex RT-qPCR assay used by the diagnostic laboratories. Of this group of specimens, n=7/16 also gave a positive fluorescent LAMP signal, while n=9/16 failed to signal above baseline in the 30 minute assay time. Fluorescent RT-LAMP is measured by the time in minutes required to generate fluorescence signal (ΔR) over baseline in the assay. Results Section 

5. Authors should show the easily understandable interpretation of Table 1-4.

If they discuss sensitivity, specificity, and false positives, a 2×2 table is easier to understand. If they discuss the viral load, Table 1-4 is indispensable.

• We revised the Tables and provide an explanation of results in the revised manuscript. Of the original SARS-CoV-2 positive (n=50) NP samples, 84% (n=42/50) were true N gene positives as determined by our RT-qPCR assay. Fluorescent RT-LAMP assay results were 98% (n=41/42) concordant on these same true N gene positive specimens. Eight SARS-CoV-2 positive NP specimens, as determined by the UGA diagnostic laboratories, failed to amplify the N gene target in our RT-qPCR assay. However, the internal control amplicon (human RNAseP) amplified competently in all NP specimens we tested indicating that extracted RNA was intact. N=7/8 of the failed RT-qPCR NP specimens resulted in a positive fluorescent RT-LAMP assay discordant with our RT-qPCR result for these NP specimens, but in agreement with the original SARS-CoV-2 positive status confirmed by the diagnostic labs. This could suggest that sample preparation differences pursuant to running the assays (RNA purification for RT-qPCR or minimal pre-processing for RT-LAMP) contributes to different results. We show that the fluorescent RT-LAMP assay sensitivity is more than sufficient to reliably detect SARS-CoV-2 in NP samples as the viral load in SARS-CoV-2 patients is reported to range from 641 copies/mL to 1011 copies/mL with a median of 104 - 105 [31]. Of the original group of SARS-CoV-2 negative (n=84) NP samples, 81% (n=68/84) were true N gene negatives as determined by our RT-qPCR assay. Fluorescent RT-LAMP assay results were 87% (n=59/68) concordant on these same true N gene negative specimens. Sixteen SARS-CoV-2 negative NP specimens, as determined by the diagnostic laboratories, gave a positive result in our RT-qPCR assay. These results could represent RT-qPCR false positives in our assay, but were clearly determined SARS-COV-2 negative by the diagnostic labs with the difference likely owing to their broad-specificity RT-qPCR assay. Fluorescent RT-LAMP, with regards to the true N gene negative NP specimens, gave 13% (n=9/68) with positive results discordant with RT-qPCR. Discussion section 

6. Authors should avoid including the discussions in the result.

・The viral load in SARS-CoV-2 patients is reported to range from 641 copies/mL to 1011 copies/mL with a median of 104 - 105 [31].

・Form ‘Importantly’ to ‘when the Ct value was <30 and >40 for negative fluorescent RT-LAMP result’.

• This has been corrected. Statement amended and moved to the Discussion section. This statement is amended in Discussion section to provide context within the RT- LAMP reaction results. We show that the fluorescent RT-LAMP assay sensitivity is more than sufficient to reliably detect SARS-CoV-2 in NP samples with respect to virus load, as SARS-CoV-2 patients have been reported to have a range from 641 copies/mL to 1011 copies/mL with a median of 104 - 105 [31]. 

Minor comments are as follows:

7. The title of Figure 2 is the same as that of figure 3. Authors should set appropriate titles.

• This has been corrected to: Figure 3. SARS-CoV-2 Live Virus Fluorescent RT-LAMP Assay 

8. In the legend of figure 3. A, B between ‘and’ and ‘Fluorescent RT-LAMP assay’ is a typo?

• This has been corrected in Figure 3 legend. 

9. Authors should confirm the titles of Figure 4.

The title of A is SARS CoV-2 WA-1 Cell culture lysate?

The title of B is SARS CoV-2 Variant B. 1. 1. 7 variant cell culture lysate?

• Corrected Figure 4 titles and sub-titles

• Figure 4. SARS-CoV-2 Live Virus CDC EUA N1 RT-qPCR Assay

• SARS-CoV-2 WA-1 Cell Culture Lysate

• SARS-CoV-2 Variant - B.1.1.7 (alpha) (20I/501Y.V1) Cell Culture Lysate

---

## [Decision Letter · Decision Letter 1]

26 Jul 2021

PONE-D-21-13279R1

Isothermal Amplification and Fluorescent Detection of SARS-CoV-2 and SARS-CoV-2 Variant Virus in Nasopharyngeal Swabs

PLOS ONE

Dear Dr. Tripp,

Thank you for submitting your manuscript to PLOS ONE. After careful consideration, we feel that it has merit but does not fully meet PLOS ONE’s publication criteria as it currently stands. Therefore, we invite you to submit a revised version of the manuscript that addresses the points raised during the review process.

Please describe about the IRB as the reviewer suggested. At the next round of review I will judge "accept or reject" directly. 

We look forward to receiving your revised manuscript.

Kind regards,

Etsuro Ito

Academic Editor

PLOS ONE

Reviewers' comments:

Reviewer's Responses to Questions

**Comments to the Author**

1. If the authors have adequately addressed your comments raised in a previous round of review and you feel that this manuscript is now acceptable for publication, you may indicate that here to bypass the “Comments to the Author” section, enter your conflict of interest statement in the “Confidential to Editor” section, and submit your "Accept" recommendation.

Reviewer #3: All comments have been addressed

Reviewer #4: All comments have been addressed

2. Is the manuscript technically sound, and do the data support the conclusions?

Reviewer #3: Yes

Reviewer #4: Yes

3. Has the statistical analysis been performed appropriately and rigorously? 

Reviewer #3: Yes

Reviewer #4: No

4. Have the authors made all data underlying the findings in their manuscript fully available?

Reviewer #3: Yes

Reviewer #4: Yes

5. Is the manuscript presented in an intelligible fashion and written in standard English?

Reviewer #3: Yes

Reviewer #4: Yes

6. Review Comments to the Author

Reviewer #3: All comments have been addressed by the authors. I have no other further comments. Well done and keep it up!

Reviewer #4: The authors appropriately responded to my comments. However, this article still possesses some technical short comings and requires considerable revision as listed below.

Major comments are as follows:

1. This study lacks IRB approval because of using anonymous data. However, authors obtained written informed consent from the participants. In what format did they obtain the consent? Who checked the format? This is very important as a premise of research.

2. Detection methods usually evaluate the accuracy using sensitivity and specificity. Table 1 and 2 are not according to the theory and difficult to understand. Readers may not understand why authors considered previous positive samples (n=50) and negative samples (n=84) separately. It may be easy to understand the results when they evaluate the accuracy in 134 samples referring to qPCR or fluorescent RT-LAMP as follow. Then they can calculate sensitivity and specificity of RT-LAMP.

　　　　　　　　　 ＱPCR

　　　　　　 Positive Negative Total

　　　　Positive

LAMP　Negative

　　　　Total 　　　　　　 134

7. PLOS authors have the option to publish the peer review history of their article (what does this mean?). If published, this will include your full peer review and any attached files.

Reviewer #3: **Yes: **Lau Yee Ling

Reviewer #4: No

---

## [Author Response · Author response to Decision Letter 1]

13 Aug 2021

Response to Reviewer

Reviewer #3: All comments have been addressed by the authors. I have no other further comments. Well done and keep it up!

Reviewer #4: The authors appropriately responded to my comments. However, this article still possesses some technical short comings and requires considerable revision as listed below.

Major comments are as follows:

1. This study lacks IRB approval because of using anonymous data. However, authors obtained written informed consent from the participants. In what format did they obtain the consent? Who checked the format? This is very important as a premise of research.

• As stated on pages 269-273 in the revised manuscript, “Deidentified samples were collected from consenting adult volunteers by the Georgia National Guard or the University of Georgia (UGA) COVID-19 Task Force following written approval and were tested from frozen -80°C samples collected by the Georgia Taskforce for COVID-19, the Georgia National Guard, or the University of Georgia Diagnostic Laboratories.” A copy of the consent form is now attached in the Plos One editorial manager. 

• Due to the pandemic ‘community surveillance’ was instituted across the state of Georgia and the UGA campus. The program consisted of collecting nasal swabs from the volunteer population in order to determine the incidence of COVID-19. The results of such surveillance was aggregate, population-level data to provide leaders with a high-level understanding of the effects of COVID-19 on the health and safety of areas of the state and the campus. The COVID-19 Task force included diagnostic testing by a CLIA-approved laboratories at UGA. Volunteers with a positive result were encouraged to seek healthcare.

2. Detection methods usually evaluate the accuracy using sensitivity and specificity. Table 1 and 2 are not according to the theory and difficult to understand. Readers may not understand why authors considered previous positive samples (n=50) and negative samples (n=84) separately. It may be easy to understand the results when they evaluate the accuracy in 134 samples referring to qPCR or fluorescent RT-LAMP as follow. Then they can calculate sensitivity and specificity of RT-LAMP.

• We regret any misunderstanding and reiterate as we noted in the manuscript that the study compares RT-LAMP detection of SARS-CoV-2 to RT-qPCR detection from the same patient NP samples. We examined both SARS-CoV-2 positive (n=50) and negative (n=84) patient NP samples that were previously confirmed by qualified RT-qPCR assay from the UGA diagnostic laboratories. We explain in the manuscript that the specificity of the state laboratory RT-qPCR assay is multiplex for SARS-CoV-2 detection, in contrast to our N gene specific RT-LAMP assay. For this reason we employ the SARS-CoV-2 N gene specific CDC EUA N1 RT-qPCR assay for comparison of results to that of our RT-LAMP assay on the same NP samples. The RT-LAMP assay protocol uses minimally-processed NP sample as template in the reaction, as stated in the Methods section. In contrast, the RT-qPCR protocol uses conventionally purified RNA as template. 

• To reiterate, as we show in the manuscript, we evaluated NP samples to determine if the RT-LAMP assay would produce results similar to RT-qPCR to help explain the concordance and discordance reported in Tables 1 and 2. We believe it is important to note that in the positive sample group we were unable to detect SARS-CoV-2 N gene by RT-qPCR in 8/50 samples. This result accounts for most of the discordant results with RT-LAMP in the positive sample group. Similarly, in the negative sample group, we detected SARS-CoV-2 N gene in 16/84 samples by RT-qPCR. We believe it is important to report that this accounts for half of the total discordant results (18/84) in the negative NP sample group. RT-qPCR vs RT-LAMP results for the two sample groups combined: N=134 total samples, 108 (81%) in concordance and 26(19%) in discordance as is discussed in the manuscript.

• To recap, we show in the manuscript the direct comparison of RT-LAMP sensitivity for SARS-CoV-2 detection to RT-qPCR using standardized templates as shown for SARS-CoV-2 N gene synthetic RNA in Figures 1 and 2, and in Supplemental Figure 1 using purified genomic viral RNA as template in both assays. Assay specificity is reported using cultured SARS-CoV-2 wild type and Alpha variant viruses, and the closely related circulating human coronaviruses as templates in RT-LAMP and RT-qPCR in Figures 3 and 4.

---

## [Editor Report · Decision Letter 2]

6 Sep 2021

Isothermal Amplification and Fluorescent Detection of SARS-CoV-2 and SARS-CoV-2 Variant Virus in Nasopharyngeal Swabs

PONE-D-21-13279R2

Dear Dr. Tripp,

We’re pleased to inform you that your manuscript has been judged scientifically suitable for publication and will be formally accepted for publication once it meets all outstanding technical requirements.

Kind regards,

Etsuro Ito

Academic Editor

PLOS ONE

---

## [Editor Report · Acceptance letter]

9 Sep 2021

PONE-D-21-13279R2 

Isothermal Amplification and Fluorescent Detection of SARS-CoV-2 and SARS-CoV-2 Variant Virus in Nasopharyngeal Swabs 

Dear Dr. Tripp:

I'm pleased to inform you that your manuscript has been deemed suitable for publication in PLOS ONE. Congratulations! Your manuscript is now with our production department. 

Kind regards, 

on behalf of

Prof. Etsuro Ito 

Academic Editor

PLOS ONE